# NFATc1 controls the cytotoxicity of CD8+ T cells

Stefan Klein-Hessling[1], Khalid Muhammad[1], Matthias Klein[2], Tobias Pusch[1], Ronald Rudolf[1], Jessica Flöter[3], Musga Qureischi[4], Andreas Beilhack[4], Martin Vaeth[1,8], Carsten Kummerow [5], Christian Backes[5], Rouven Schoppmeyer[5], Ulrike Hahn[6], Markus Hoth [5], Tobias Bopp[2], Friederike Berberich-Siebelt[1], Amiya Patra[1,9], Andris Avots[1], Nora Müller[7], Almut Schulze[3] & Edgar Serfling[1]

Cytotoxic T lymphocytes are effector CD8+ T cells that eradicate infected and malignant cells. Here we show that the transcription factor NFATc1 controls the cytotoxicity of mouse cytotoxic T lymphocytes. Activation of $Nfatc1^{-/-}$ cytotoxic T lymphocytes showed a defective cytoskeleton organization and recruitment of cytosolic organelles to immunological synapses. These cells have reduced cytotoxicity against tumor cells, and mice with NFATc1-deficient T cells are defective in controlling Listeria infection. Transcriptome analysis shows diminished RNA levels of numerous genes in $Nfatc1^{-/-}$ CD8+ T cells, including $Tbx21$, $Gzmb$ and genes encoding cytokines and chemokines, and genes controlling glycolysis. $Nfatc1^{-/-}$, but not $Nfatc2^{-/-}$ CD8+ T cells have an impaired metabolic switch to glycolysis, which can be restored by IL-2. Genome-wide ChIP-seq shows that NFATc1 binds many genes that control cytotoxic T lymphocyte activity. Together these data indicate that NFATc1 is an important regulator of cytotoxic T lymphocyte effector functions.

[1] Department of Molecular Pathology, Institute of Pathology, and Comprehensive Cancer Center Mainfranken, University of Würzburg, D-97080 Würzburg, Germany. [2] Institute for Immunology, University Medical Center, University of Mainz, D-55131 Mainz, Germany. [3] Department of Biochemistry and Molecular Biology, Theodor-Boveri-Institute, Biocenter, and Comprehensive Cancer Center Mainfranken, University of Würzburg, D-97074 Würzburg, Germany. [4] Department of Medicine II, Würzburg University Hospital, Research Center for Infectious Diseases, and Interdisciplinary Center for Clinical Science Research, University Würzburg, D-97078 Würzburg, Germany. [5] Department of Biophysics, CIPMM, Faculty of Medicine, Saarland University, D-66421 Homburg, Germany. [6] Cellular Neurophysiology, CIPMM, Saarland University, D-66421 Homburg, Germany. [7] Institute for Virology and Immunobiology, University of Würzburg, D-97080 Würzburg, Germany. [8] Present address: Department of Pathology, New York University School of Medicine, 10016 New York, USA. [9] Present address: Institute of Translational and Stratified Medicine, Peninsula Schools of Medicine and Dentistry, University of Plymouth, Plymouth PL6 8BU, UK. Stefan Klein-Hessling, Khalid Muhammad, Matthias Klein and Tobias Pusch contributed equally to this work. Correspondence and requests for materials should be addressed to E.S. (email: serfling.e@mail.uni-wuerzburg.de)

The primary function of CD8+ T cells is to eradicate infected and tumor cells. Upon activation and differentiation of naïve CD8+ T cells to effector CD8+ T cells, cytotoxic T lymphocytes (CTL) synthesize large amounts of the inflammatory cytokines IFNγ and TNF, and the cytotoxic effector molecules perforin and granzyme B, which are deposited in lytic granules in the cytosol. Upon contact of CTLs with target cells, the lytic granules are re-orientated and recruited to the immunological synapse (IS), along with the microtubule-organizing center (MTOC), the Golgi apparatus and mitochondria[1, 2]. At or near the immunological synapse, lytic granules fuse with the cell membrane and release perforins and granzymes to kill target cells[3].

CD8+ T cell contact with cognate antigen leads to intracellular T cell receptor (TCR)-mediated signaling that, along with co-stimulatory signals, orchestrates gene expression programs to control the expansion and differentiation of CD8+ T cells to CTLs in peripheral lymphoid organs. Upon primary stimulation and the generation of effector cells, most of the activated CD8+ T cells die, but a small number of cells survive and develop into memory CD8+ T cells. According to surface expression and similar to CD4+ T cells, memory CD8+ T cells are classified into central memory CD8+ $T_{CM}$ cells and effector memory CD8+ $T_{EM}$ cells that differ in their homing capacity and effector function[4, 5]. However, the identification of tissue-resident memory $T_{RM}$ cell subsets suggests that a variety of other CD8+ memory T cells exist to ensure optimal immunity against infection and cancer[6].

One prominent signaling network that has an important function in the generation and function of activated CD8+ T cells and CTLs is the Ca++/calcineurin/NFAT network. Activation of this network is initiated by the TCR-mediated release of Ca++ from endoplasmic stores, resulting in the multimerization of Stromal interaction molecules (STIM) that contact pore-forming ORAI proteins and activate Ca++ influx from the extracellular space through Ca++ release activated Ca++ channels (CRAC)[7]. The rise of intracellular Ca++ leads to the rapid activation of the Ser/Thr-specific phosphatase calcineurin that binds and dephosphorylates the highly phosphorylated cytosolic NFAT proteins, and stimulates their nuclear import[8].

The family of NFAT transcription factors consists of five members that share a common DNA-binding domain of approximately 300 amino acid residues. There are only a few studies on NFAT transcription factors in CD8+ T cells. In one study, a defective nuclear translocation of NFATc1 has been described for NFATc1 in CD8+ T cells upon chronic infection[9], whereas in another study a predominant nuclear localization of NFATc1 was reported for anergic CD8+ T cells[10]. The effect of NFATc1 (NFAT2) ablation on CD8+ T cell physiology has been reported[11], but genome-wide assays on the effect of NFATc1 on gene expression in CTLs have not.

Here we show that upon TCR stimulation, ablation of NFATc1 results in an impaired formation of F-actin rings around the immunological synapse in CTLs, and poor recruitment of lytic granules and mitochondria to the synapse. Genome-wide transcriptome and chromatin immuno precipitation (ChIP) assays show that NFATc1 controls genes (including *Il2* and *Ifng*) that orchestrate the activity of activated CD8+ T cells (aCD8+ T cells) and CTLs. Although NFATc1 and NFATc2 overlap in their binding to most of these genes, in most cases NFATc1 has a much stronger transcriptional effect than NFATc2 in aCD8+ T cells and CTLs. Upon activation, *Nfatc1*−/− aCD8+ T cells reduce the metabolic switch from oxidative phosphorylation (OXPHOS) to glycolysis, an effect that can be restored by IL-2. Taken together our data demonstrate that NFATc1 controls transcription of genes that direct the cytotoxicity of CD8+ T cells.

## Results

**NFATc1 for cytoskeleton reorganization in activated CTL.** NFATc1 ablation alters the shape of CTLs. When WT CTLs adhere to a glass slide on which αCD3/CD28 have been attached they spread out in a velvet-like lamellipodium. By contrast, *Nfatc1*−/− CTLs form numerous spikes and filopodia-like structures which differ conspicuously from the shape of WT CTLs (Fig. 1a), and the area covered by *Nfatc1*−/− CTLs is reduced, compared to WT CTLs (Fig. 1b).

The rapid formation of IS at the TCR is accompanied by the depletion of cortical F-actin, the accumulation of MTOC, and the recruitment of mitochondria and lytic granules[12]. When we studied the formation of F-actin rings generated around the IS upon activation by total internal reflection fluorescence (TIRF) microscopy, we detected radially symmetric F-actin rings for WT cells whereas *Nfatc1*−/− CTLs generated no or smaller and irregular rings (Fig. 1b–d).

The decrease of actin density in the center of IS is accompanied by the appearance of MTOC (Fig. 1d, e) consisting of microtubules and further cytosolic components[13], including attached mitochondria and lytic granules[14]. Using TIRF we did not observe in *Nfatc1*−/− CTLs a marked delay in MTOC accumulation (Fig. 1d). However, a defect was detected in the assembly of mitochondria and lytic granules at the IS in living cells using fluorescence microscopy. Upon staining of WT and *Nfatc1*−/− CTLs for mitochondria with MitoTracker® and lytic granules with LysoTracker® followed by incubation of cells with αCD3/CD28-coated beads we detected the recruitment of mitochondria and cytotoxic granules to the IS in living WT CTLs within 1-5 min. In *Nfatc1*−/− CTLs both types of organelles did not, or slowly accumulate at the IS (Fig. 1f and Supplementary Fig. 1). Staining of CTLs with Abs directed against granzyme B revealed an accumulation of granzyme B-positive spots in the cytosol of *Nfatc1*−/− and *Nfatc1*−/− + *Nfatc2*−/− but not in WT or *Nfatc2*−/− cells, and in the supernatant of *Nfatc1*−/− CTLs a reduction in granzyme B levels was detected (Supplementary Fig. 2a, b).

**NFATc1 affects CTL-mediated killing.** The changes in morphology of *Nfatc1*−/− CTLs suggest defects in their cytolytic activity. To identify such defects at the cellular level, we investigated first the killing of MOPC 315 plasmacytoma cells expressing a luciferase reporter gene by *Nfatc1*−/− aCD8+ T cells. A marked decrease in cytolytic effector function of *Nfatc1*−/− cells was observed reaching approximately 50% of that of WT aCD8+ Ts, whereas only a slight difference was detected between *Nfatc2*−/− and WT aCD8+ Ts (Fig. 2a). A decrease in killing capacity was also observed for sorted CD62L−CD44+ $CTL_{EM}$ cells (Supplementary Fig. 3a). WT and *Nfatc1*−/− CTLs generated from CD8+ T cells of C57/BL6 mice by co-culture with irradiated Balb/c splenocytes for 6 d and A20J tumor cells as targets (see Supplementary Fig. 3b) revealed a 40% decrease in killing, compared to WT CTLs. Those *Nfatc1*−/− CTLs showed also a reduction in the surface expression of the degranulation marker CD107a whose induction is correlated with the cytotoxicity of cells (Fig. 2b). Moreover, they showed a reduction in the number of cells producing IFNγ, IL-2 and TNF whereas the number of CTLs expressing IL-17 and granzyme B was increased (Fig. 2c; Supplementary Fig. 3c). No or subtle differences were detected between WT and *Nfatc1*−/− CTLs regarding the surface expression of CD62L, CD44, CD25, FasL and CD69 (Supplementary Fig. 3d).

*Listeria monocytogenes* infection induces a strong CD8+ T cell response that combats infection[15]. To investigate the role of NFATc1 in CTL responses against *Listeria monocytogenes*,

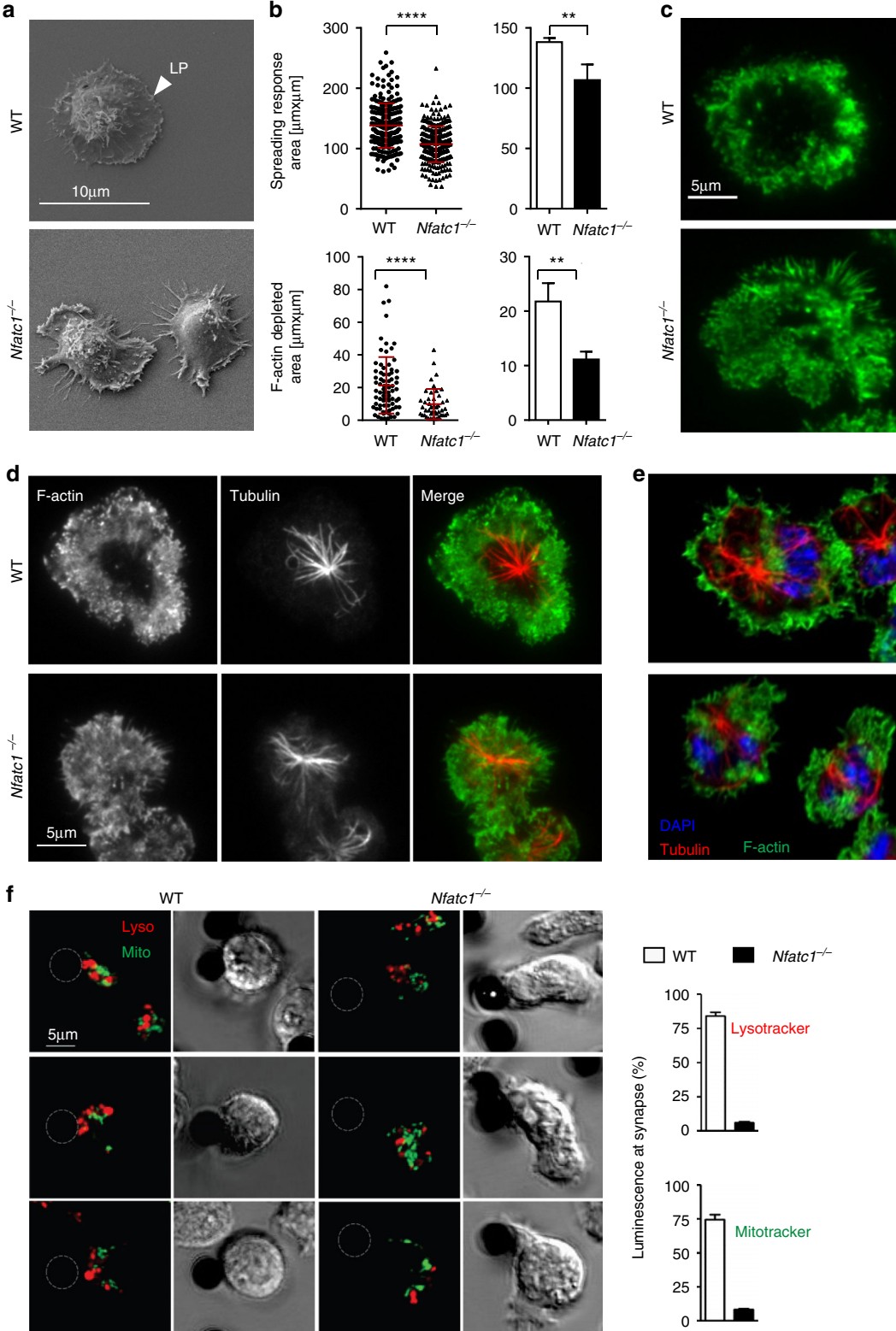

**Fig. 1** NFATc1 controls the re-organization of cytoskeleton and polarization of cytosolic organelles. **a** Spreading of WT and *Nfatc1*$^{-/-}$ (*Nfatc1*$^{f/f}$ x CD4-cre mice) CTLs on αCD3/CD28-coated glass slides within 5 min. Scanning electron microscopy. LP, lamellipodium. **b** Quantification of data of spreading (above) and F-actin ring formation (below) of CTLs activated for 5 min (*t*-test, unpaired, *p*-value < 0.0001). *Left*; each *dot/triangle* corresponds to one cell. *Right*; mean of three independent experiments. **c** Formation of F-actin rings in WT and *Nfatc1*$^{-/-}$ CTLs upon αCD3/CD28 activation within 5 min. TIRF microscopy. **d** Appearance of MTOC-tubulin near the IS of CTLs upon activation for 5 min. TIRF microscopy. **e** Organization of F-actin and MTOC-tubulin in CTLs upon activation for 20 min. Confocal microscopy. For *scale bar* see **d**. **f** Repositioning of mitochondria and lytic granules to the IS in living WT but not *Nfatc1*$^{-/-}$ CTLs. *Left*; analysis by confocal microscopy; overlay of bright field and merged fluorescence images of lysosomes (*red*) and mitochondria (*green*). The *dark dots* correspond to beads coated with αCD3/CD28. *Right*; quantification by epifluorescence microscopy. The column presentation compiles the data of more than 15 cells from 3 mice each

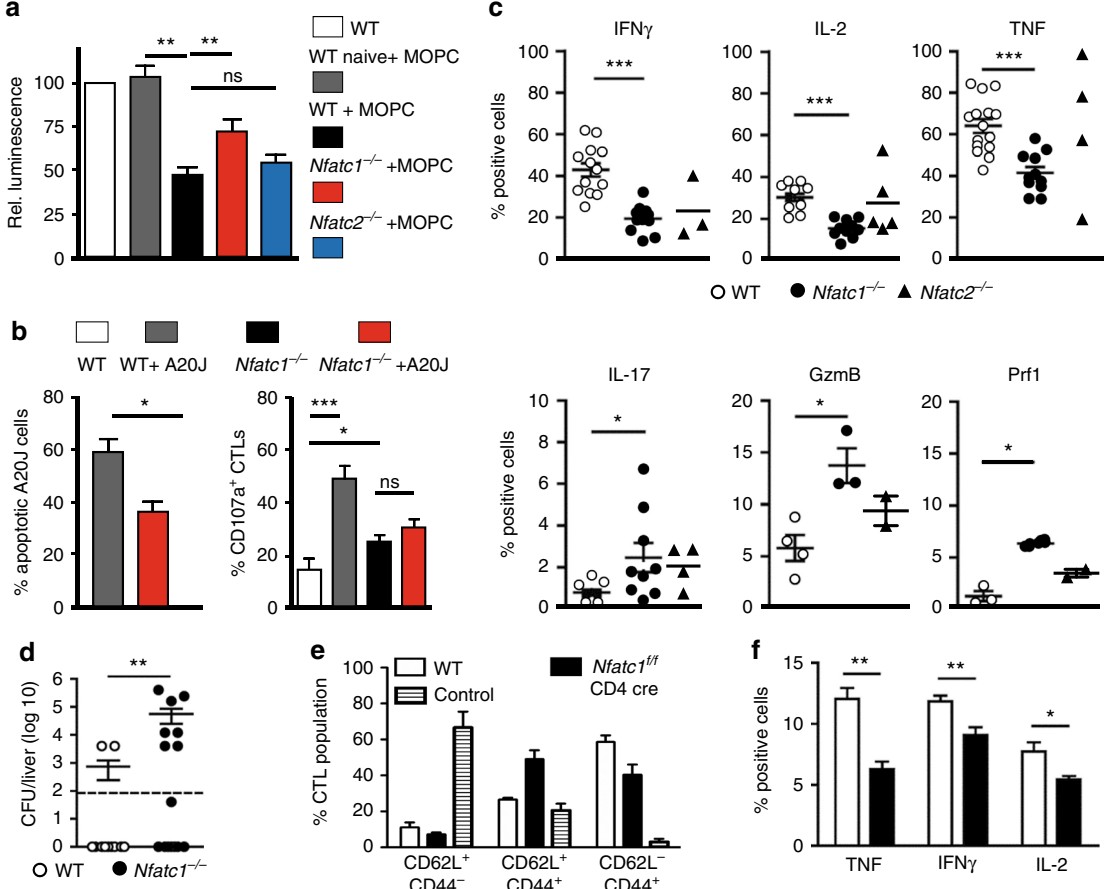

**Fig. 2** Defective killing activity of NFATc1-deficient aCD8+ Ts and CTLs. **a** Defective killing of MOPC 315 plasmacytoma cells expressing a luciferase indicator gene by aCD8+ Ts generated upon activation by αCD3/CD28 for 3 d in vitro. WT, *Nfatc1−/−* (*Nfatc1f/f* x CD4-cre mice) and *Nfatc2−/−* aCD8+ Ts were incubated with MOPC target cells for 24 h. As controls, naive CD8+ T cells were incubated, and WT aCD8+ Ts were incubated without MOPC target cells, and WT aCD8+ Ts were incubated without MOPC target cells. **b** Defective killing of A20J target cells by *Nfatc1−/−* CTLs. WT and *Nfatc1−/−* C57/B6 CTLs were generated by incubation with irradiated splenocytes from Balb/c mice for 6 d, followed incubation with A20J Balb/c tumor cells for 2 h (see Supplementary Fig. 3b for details). Left, apoptosis induction of A20J cells is presented. Right, the degranulation of CTLs is shown. **c** Changes in number of NFATc1-deficient CTLs expressing various cytokines, granzyme B (*Gzmb*) or perforin (*Prf1*). Intracellular cytokine expression of CTLs was measured by flow cytometry upon stimulation by T + I for 6 h. Granzyme B and perforin expression was measured without T + I stimulation. **d** Defective clearance of *Listeria monocytogenes* upon infection in mice bearing NFATc1-deficient T cells. WT and *Nfatc1f/f* x CD4-cre mice were infected with 5 × 10⁵ CFU ΔActA Lm-Ova[66] bacteria. 5 d later, the mice were sacrificed and *Lm* titers in livers were determined. **e** Distribution of splenic CTLs upon *Lm* infection of WT mice and *Nfatc1 f/f* x CD4-cre mice. Control, uninfected mice. **f** Number of cytokine-producing CTLs upon *Lm* infection in *Nfatc1f/f* x CD4-cre mice. Data are shown as means ± SEM. Unpaired Student's t-test was used for statistics

we infected WT and *Nfatc1f/f* x CD4-cre mice with the transgenic (tg) strain ΔActA *Lm*-Ova lacking protein ActA expression that induces actin assembly[16]. *Listeria monocytogenes* bacteria were detected in the livers of 8 of the 14 *Nfatc1f/f* x CD4-cre mice infected with ΔActA *Lm*-Ova for 5 d whereas only two of the WT mice showed signs of infection (Fig. 2d). Infected *Nfatc1fl/fl* x CD4-cre mice showed an increase in number of CD62L+, CD44+ central and decrease of CD62L−, CD44+ effector memory cells, and a reduced number of cells producing TNF and IL-2, compared to WT mice (Fig. 2e, f; Supplementary Fig. 3e).

**NFAT expression in aCD8+ T cells and CTLs**. The effect of NFATc1 on CTL activity led us to study NFAT expression in CD8+ T cell cultures. To generate CTLs, splenic CD8+ T cells were stimulated first with plate-bound αCD3/CD28 for 3 d (resulting in aCD8+ T cells), followed by incubation with IL-2 for 5–6 d. To mimic contacts with target cells, finally CTLs were stimulated by TPA and Ionomycin (T + I) for 5 h (Fig. 3a). These

culture conditions approximate CD8+ T cell responses in vivo and allowed us to generate large numbers of cells for molecular studies.

Stimulation of CD8+ T cells by αCD3/CD28 resulted in a strong increase of NFATc1 expression within 1 d. At the RNA level, this led to a moderate 3 fold increase in P1 promoter-directed *Nfatc1* and a decrease in both P2-directed *Nfatc1* and *Nfatc2* transcripts (Fig. 3b, c). At the protein level, a strong increase in short isoform NFATc1/αA was observed (see asterisks in Fig. 3d). However, further incubation of aCD8+ Ts in IL-2 medium led to the disappearance of NFATc1/αA RNA and protein (Fig. 3c, d). Stimulation of CD8+ T cells by αCD3/CD28 resulted in a slow increase in nuclear NFATc2 that further increased upon incubation in IL-2 medium for 2-3 d and declined afterward.

T + I re-stimulation resulted in an increase of nuclear NFATc2 in CTL + cells (Fig. 3e).

CTL + cells rapidly re-gained the ability to synthesize NFATc1 proteins. However, several of the newly synthesized NFATc1 proteins were smaller than NFATc1/αA. Since incubation of

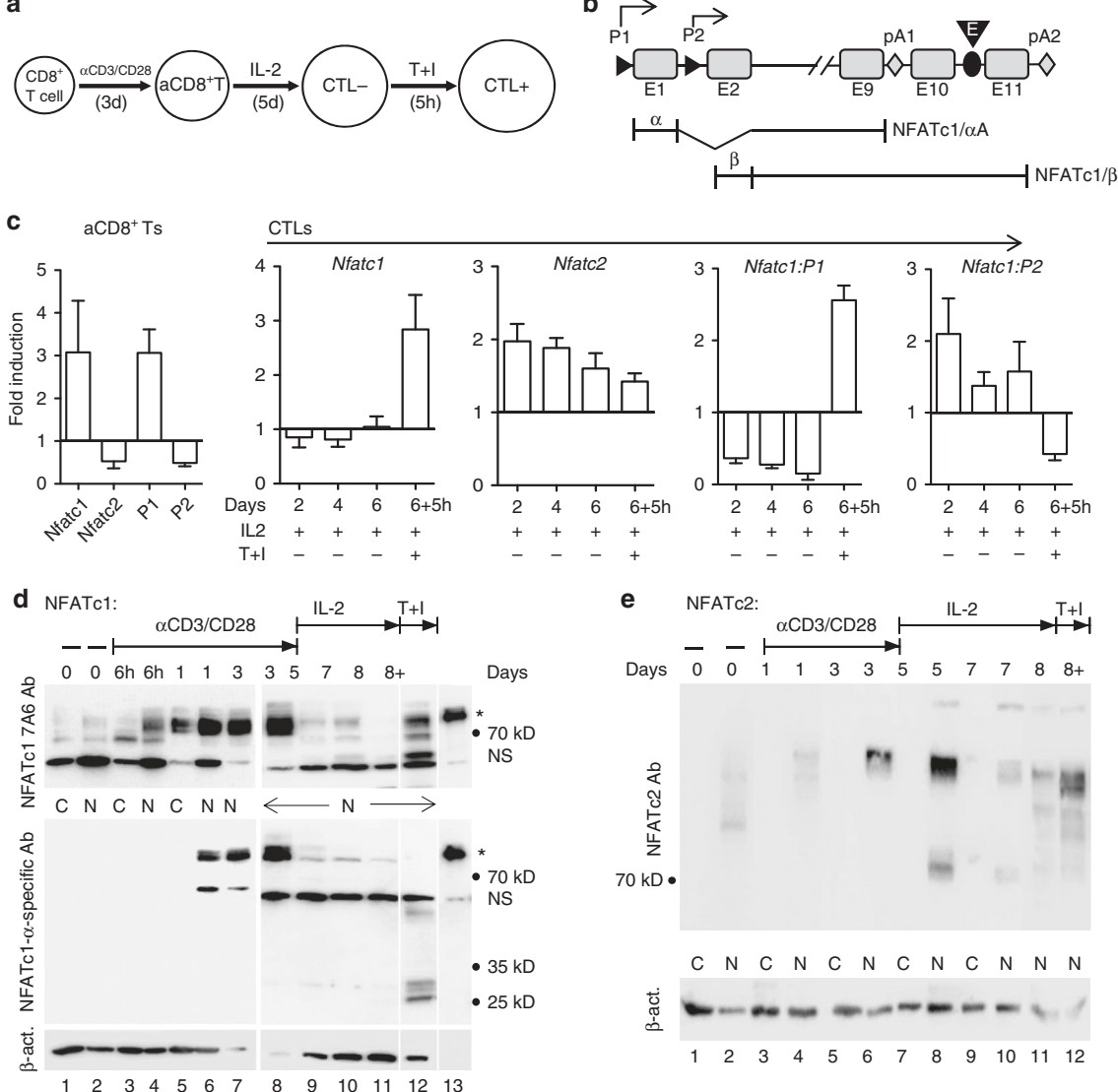

**Fig. 3** NFATc1 induction in murine CD8[+] T cells. **a** Scheme of generation of aCD8[+] Ts and CTLs from splenic CD8[+] T cells in vitro. **b** Scheme of the murine *Nfatc1* gene[67, 68]. P1, P2, promoters; pA1, pA2, polyadenylation sites; E, intronic enhancer. **c** Induction of the *Nfatc1* and *Nfatc2* genes at the transcriptional level. RT-PCR assays, normalized by Actb and relative to naïve CD8[+] T cells. Left, NFATc1 and c2 RNA levels in CD8[+] T cells, upon stimulation by αCD3/CD28 for 3 d. Right, RNA levels in CTL + cells incubated with αCD3/CD28 for 3 d followed by incubation for 2, 4 and 6 d with 100 U/ml IL-2, and for 6 d IL-2 and T + I for 5 h. **d** Immunoblot assays for the detection of NFATc1 (*upper* blot) and NFATc1/αA (*middle* blot) in aCD8[+] Ts and CTLs. As loading control, the blot was also incubated with an Ab against β-actin. CD8[+] T cells were left untreated (*lanes 1 + 2*) or treated by αCD3/CD28 for 6 h (*3 + 4*), 1 d (*5 + 6*) or 3 d (*7 + 8*). In *lanes 9-11*, aCD8[+] Ts treated with αCD3/CD28 for 3 d were maintained for 2 d (*lane 9*), 4 d (*10*) and 5 d (*11*) in medium containing 100 U/ml IL-2. In *lane 12*, in addition, CTLs were treated with T + I for 5 h. In *lanes 13*, NFATc1/αA stably expressed in KT12 cells was fractionated as marker. C, N, cytosolic and nuclear proteins. **e** Immunoblot for the detection of NFATc2. CD8[+] Ts were treated and processed as in **d**. Typical blots of more than three assays are shown

NFATc1/αA with protein extracts of CTLs resulted in peptides of the same size, they are most likely degradation products of NFATc1/αA that often lost the α-peptide (see lane 12 in Fig. 3d, and compare Supplementary Fig. 4a with Fig. 3d). As described previously[17], such cleavage was also detected for NFATc2 (Fig. 3e).

**NFAT-mediated gene expression in aCD8[+] T cells and CTLs.** To determine the transcriptional program controlled by NFATc1 in CD8[+] T cells, we performed transcriptome analysis of aCD8[+] Ts and CTLs generated from splenic CD8[+] T cells of WT and *Nfatc1*[f/f] x CD4-cre mice that did not show any NFATc1

expression in their aCD8[+] Ts and CTLs (Supplementary Fig. 4b). For comparison, we also determined the transcriptome of *Nfatc2*[−/−] mice by next generation sequencing (NGS). We sequenced mRNAs from (i) activated CD8[+]Ts (aCD8[+] Ts), (ii) CTL− and (iii) CTL + cells.

In the three types of CD8[+] T cells shown in Fig. 3a, 6000–6500 genes were expressed in more than 4 reads per kilobase per million mapped reads (RPKMs). In the heat map, transcripts of 561 genes are listed (Fig. 4a) whose expression was found to be changed 2 fold or more in at least one cell type. As expected from the high levels of nuclear NFATc1 in aCD8 + T cells and CTLs, we detected a 2.5-fold transcriptional alteration in 188 and 357 genes between WT and *Nfatc1*[−/−] aCD8 + T cells and CTLs

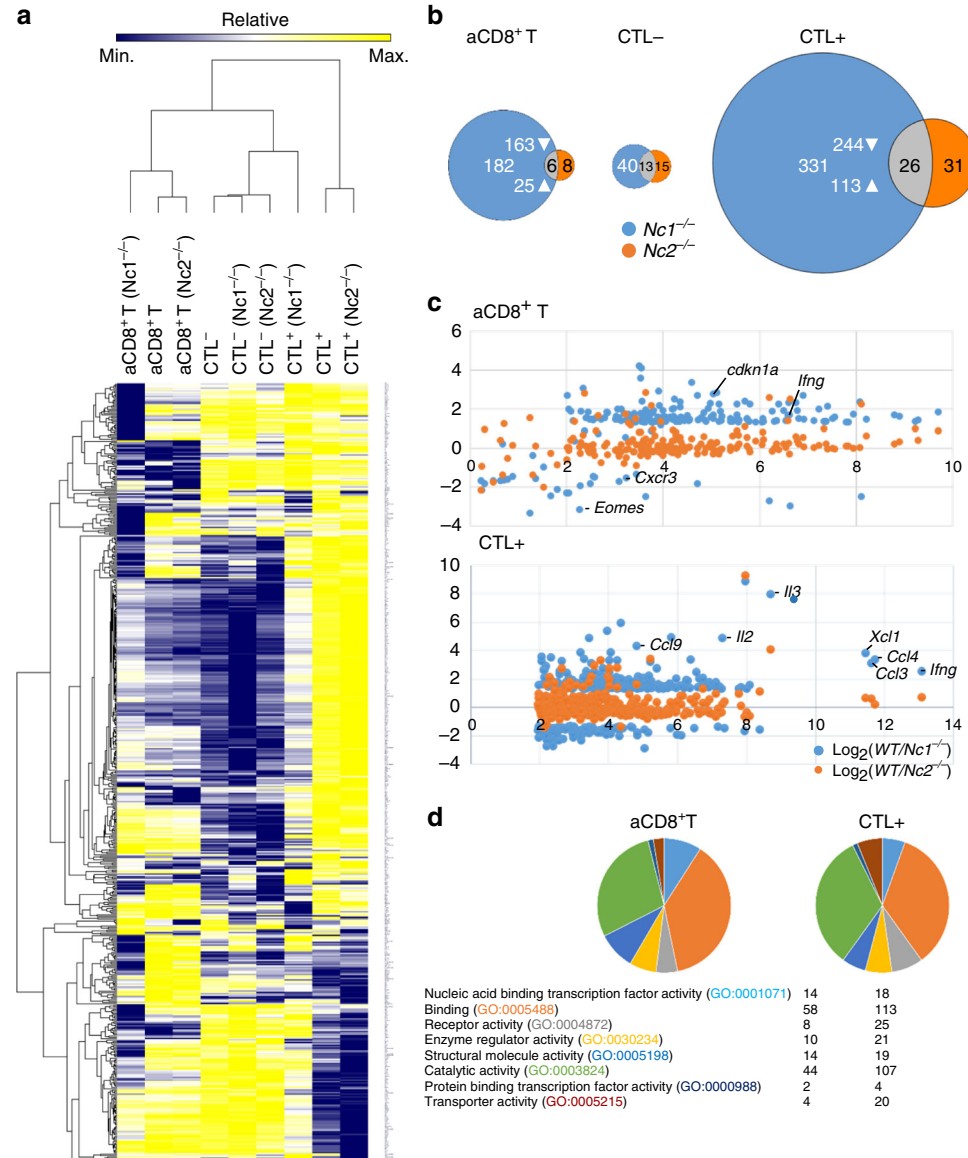

**Fig. 4** Transcriptome assays of CD8+ T cells by next generation sequencing. **a** Heat map of 561 genes whose expression changed twofold in either *Nfatc1−/−* (*Nfatc1f/f* x CD4-cre mice) or *Nfatc2−/−* aCD8+ Ts and CTLs. **b** Number of genes whose expression was changed 2.5-fold in either aCD8+ Ts, CTL− or CTL + cells. Genes whose expression was specifically changed in *Nfatc1−/−* cells are shown in *blue*, those changed in *Nfatc2−/−* cells in orange, and those changed in both in *gray*. **c** Scatterplots of NFATc1- and NFATc2-dependent genes relative to Log2WT expression in aCD8+ Ts (*upper panel*) and CTL + cells (*lower panel*). **d** Functional annotation of genes changed 2.5-fold in RNA levels in NFATc1-deficient aCD8+ Ts and CTL + cells according to the PANTHER pathway analysis[69]

respectively. Only 14 and 57 genes were changed in *Nfatc2−/−* aCD8+ Ts and CTLs + . We observed only 6 genes that were affected in both *Nfatc1−/−* and *Nfatc2−/−* aCD8+ Ts, whereas in CTLs− the transcription of 13 and in CTLs + 26 genes were changed (Fig. 4b). Among them are the *Ifng*, *Il3*, *Batf3* and *CD40l* genes, and the genes encoding the chemokines Ccl3, Ccl4 and Xcl1.

The stronger effect of NFATc1 on gene expression in aCD8+ Ts and CTLs + is also visualized in scatterplots of NFAT-dependent genes. While ablating NFATc2 exerted a relative mild effect, NFATc1 ablation resulted in much stronger changes in gene expression in aCD8+ Ts and CTLs + (Fig. 4c). Functional annotation of NFATc1-dependent genes in aCD8+ Ts and CTLs + revealed that a large proportion of deregulated genes encoding enzymes in both cell types (Fig. 4d).

To identify the chromosomal sites and genes to which NFATc1/A binds in CTLs + we performed ChIP seq assays. Since ChIP-grade NFATc1 Abs are not available, we created a BAC tg mouse line in which the biotin-ligase BirA from *E. coli* and chimeric NFATc1/A-Bio proteins are co-expressed[18] (Fig. 5a). From those mice, CD8+ T cells were isolated, differentiated into CTLs and stimulated with T + I for 5 h (to generate CTL + cells). Cross-linked NFATc1/A-chromatin complexes could be isolated with high specificity using streptavidin-coated magnetic beads and, upon removal of cross-links and DNA isolation, subjected to NGS of DNA.

Similar to the endogenous *Nfatc1* gene, the BAC *Nfatc1* tg was induced upon induction of CTLs by T + I for 5 h (Fig. 5b). The results of ChIP seq assays revealed 19 759 DNA peaks upon annotation to the mm9 mouse genome. Motif analyses within these short DNA stretches resulted in the typical TGGAAA (or—

on the reversed strand—TTTCCA) NFAT 'core' binding motif (Fig. 5c). Among the 357 genes whose RNA expression differed more than 2.5-fold between WT and $Nfatc1^{-/-}$ CTL cells stimulated by T + I for 5 h, 199 genes (56%) showed peaks for NFATc1/A-Bio. Therefore, they correspond to direct NFATc1 targets. According to published data[19], 187 of the 357 genes

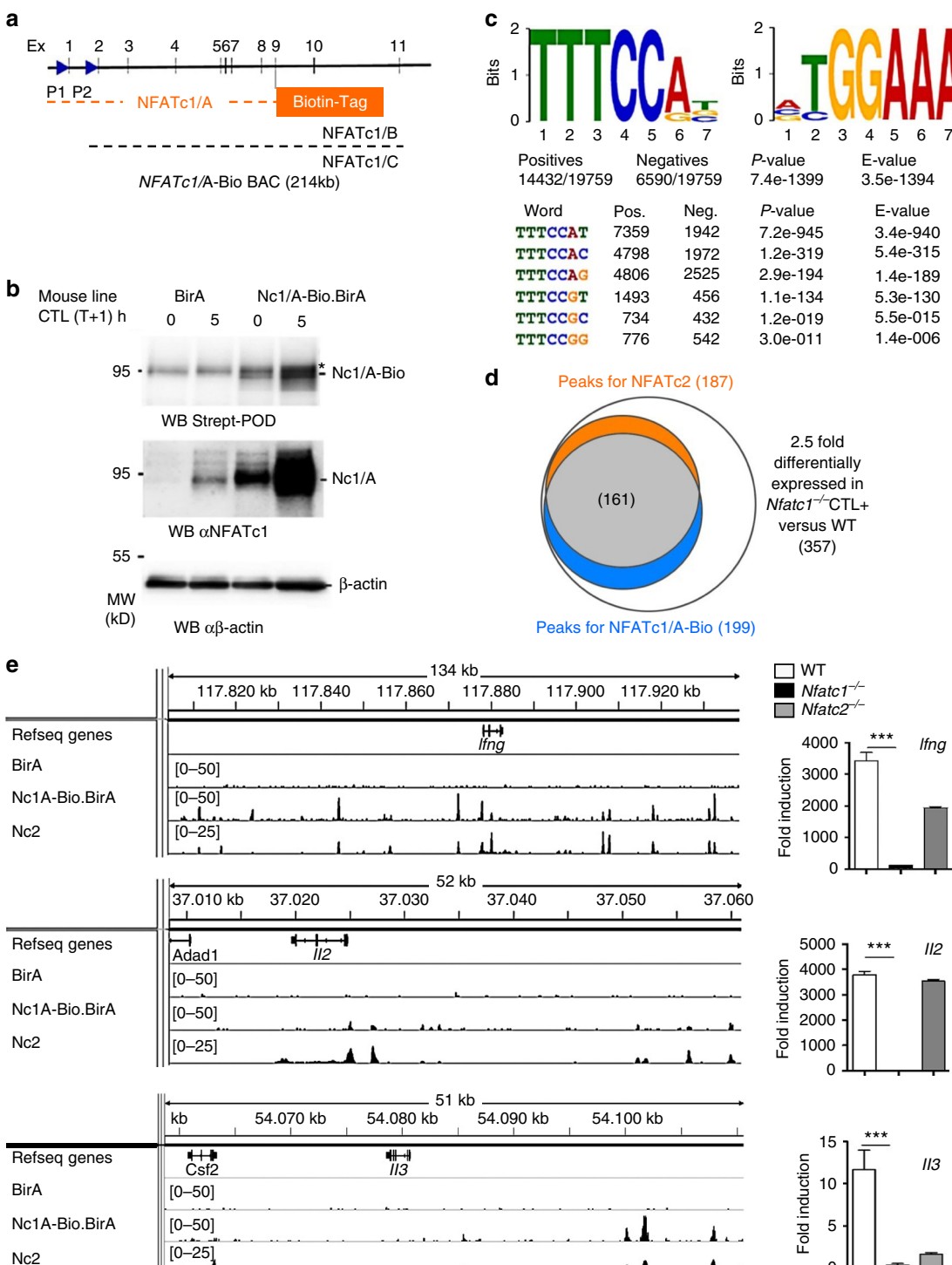

**Fig. 5** Analysis of DNA binding of NFATc1/A-Bio in CTLs by ChIP seq assays. **a** Scheme of the *Nfatc1* BAC transgene expressing tagged NFATc1/A-Bio protein in mice. The two promoters P1 and P2 and the 11 exons of the gene are indicated. **b** Immune blots of whole cell protein extracts from CTLs expressing only the *E. coli* biotin-ligase BirA, or BirA and NFATc1/A-Bio. The *upper blot* was incubated with an Ab directed against streptavidin, the *lower blot* with the NFATc1-specific mAb 7A6.The star indicates a non-specific band. **c** Motif analysis of NFATc1 binding sites in the 19 759 NFATc1/A peaks identified with a *p*-value of −5 by MACS. The letter size indicates the frequency of nucleotides within those sites (presented as 'Bits'). **d** Peaks for NFATc1/A-Bio and NFATc2[19] (*p*-value of −12 in MACS) within a distance of 100 kb to the 357 genes that were 2.5-fold differentially expressed in $Nfatc1^{-/-}$ (*Nfatc1*[f/f] x CD4-cre mice), compared to WT CTLs. 161 genes show both NFATc1/A-Bio and NFATc2 binding. **e** Binding of NFATc1/A-Bio and NFATc2[19] to the prototypical NFAT target genes *Ifng, Il2* and *Il3* in CTLs + . *Right*, real-time PCR assays of *Ifng, Il2* and *Il3* RNA levels in WT, $Nfatc1^{-/-}$ and $Nfatc2^{-/-}$ CTL + cells, normalized by Actb and relative to naïve CD8+ T cells

showed binding for NFATc2, and 161 genes showed binding for both NFATc1 and NFATc2 (Fig. 5d).

The *Ifng*, *Il2* and *Il3* genes are prototypes of NFAT target genes[20–22]. While NFATc1 and NFATc2[19] bind predominantly to the *Il2* promoter and to a site 2.3 kb upstream, both NFAT factors bind to multiple sites around the *Il3* and *Ifng* loci. NFATc1 ablation abolished RNA induction of all three genes in CTLs. Although NFATc2 binds to almost the same sites as NFATc1

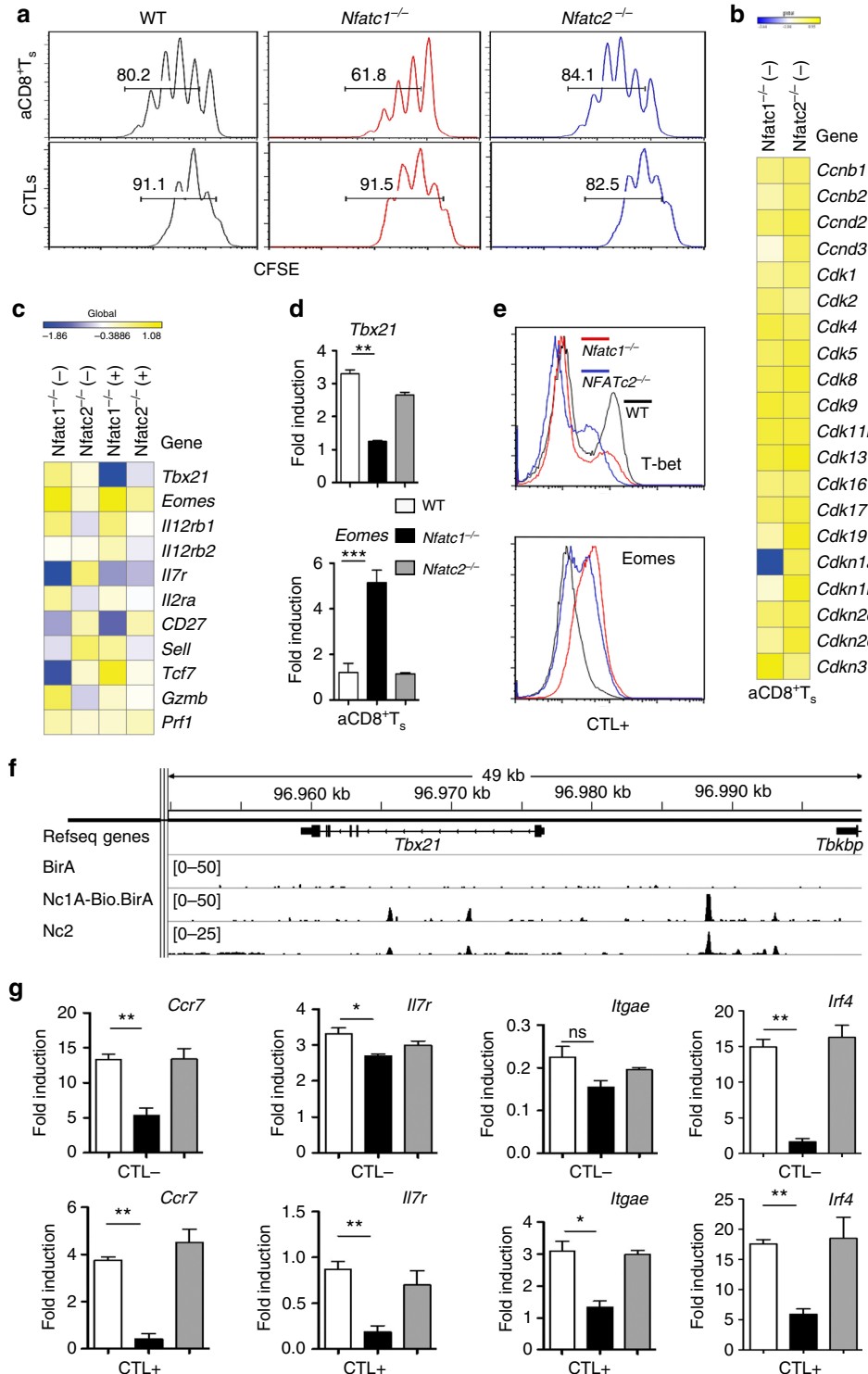

**Fig. 6** NFATc1 ablation affects the proliferation and differentiation of CD8+ T cells. **a** CFSE labeling of aCD8+ T cells stimulated for 3 d by plate-bound αCD3/CD28 (0.05/2 µg/ml), and of CTLs which were stimulated by αCD3/CD28 (5/2 µg/ml) for 3 d followed by IL-2 for 5 d. **b** Heat map of selected genes controlling the proliferation of *Nfatc1−/−* (*Nfatc1f/f* x CD4-cre mice) aCD8+ Ts. **c** Heat map of selected genes affecting the fate of CD8+ Ts. **d** Real-time PCR assays of *Tbx21* and *Eomes* RNA expression in CTL + cells, normalized by Actb and relative to naïve CD8+ T cells. **e** Flow cytometry of intracellular T-bet and Eomes expression in CTL + cells. **f** ChIP seq assay of NFATc1/A-Bio and NFATc2[19] binding to the *Tbx21* gene in CTL + cells. **g** Real-time PCR assays of *Ccr7*, *IL7r*, *Itgae* and *Irf4* RNA expression in CTLs, normalized by Actβ and relative to naïve CD8+ T cells. Data from five PCR assays are shown as means ± SEM. Unpaired Student's *t*-test was used for statistics

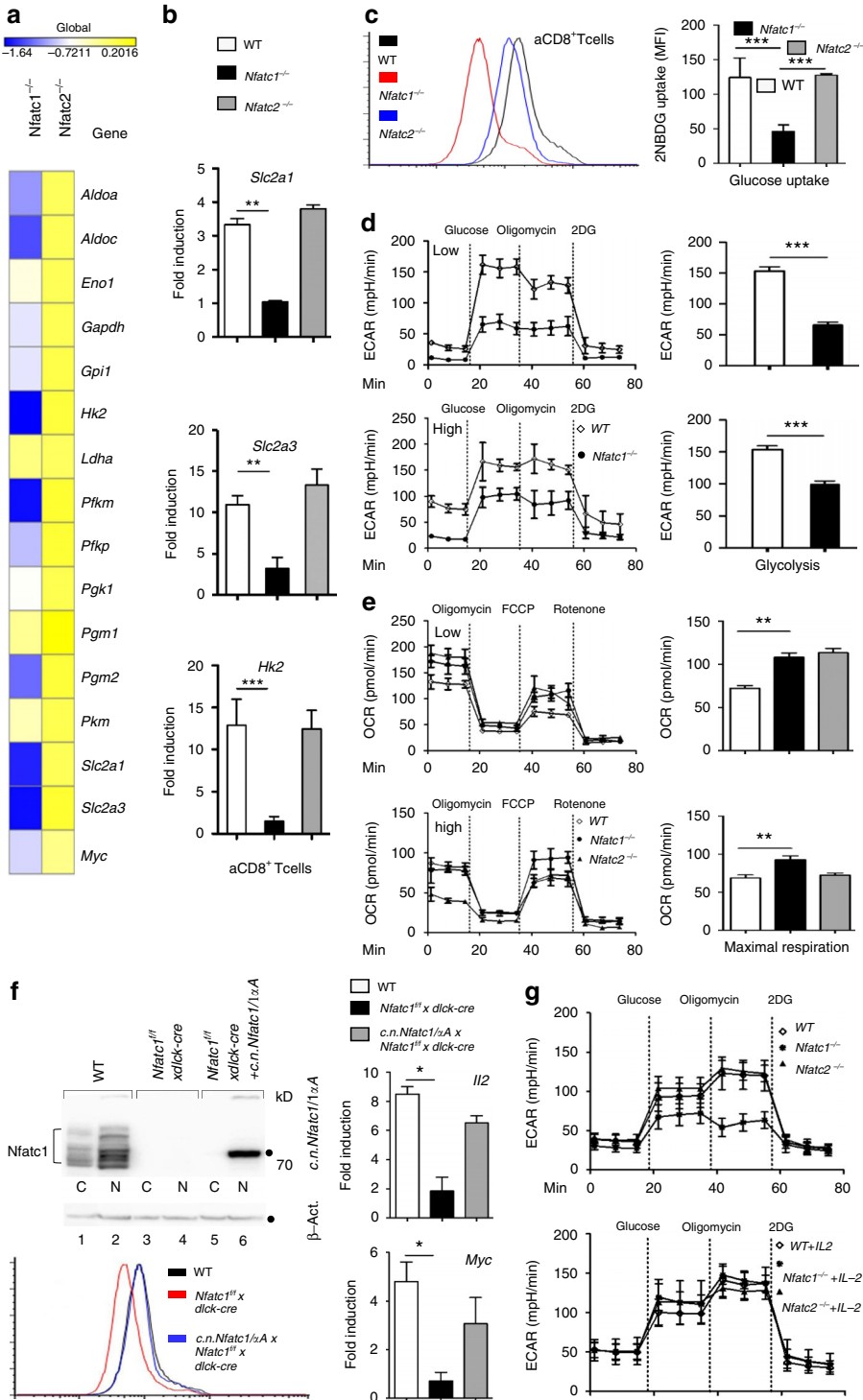

**Fig. 7** Reduced metabolic switch in *Nfatc1*[−/−] aCD8[+] T cells. **a** Heat map of RNA expression from genes encoding enzymes of glycolysis in *Nfatc1*[−/−] (*Nfatc1*[f/f] x CD4-cre mice) and *Nfatc2*[−/−] aCD8[+] Ts. **b** Real-time PCRs of *Slc2a1*, *Slc2a3* and *Hk2* RNAs encoding the glucose transporters Glut1 and Glut3, and hexokinase 2, respectively, normalized by *Actb* and relative to naïve CD8[+] cells. **c** Incorporation of 2-NBDG into aCD8[+] Ts and CTLs upon incubation for 1 h at 37 °C. Data from 4 assays were compiled. **d** Extracellular flux analysis. *Above*, 4 × 10[5] WT and *Nfatc1*[−/−] CD8[+] T cells were stimulated overnight with "low" concentrations of plate-bound αCD3 (0.05 μg/ml) and αCD28 (2 μg/ml), and subjected to extracellular flux analysis. *Below*, cells activated with "high" (standard) αCD3/CD28 concentrations (5/2 μg/ml) were assayed. **e** Mito stress test of aCD8[+]Ts treated as in **d**. **f** Below, incorporation of 2-NBDG (*above*) and extracellular flux analysis (*below*) of aCD8[+] Ts from *c.n.Nfatc1/αA x Nfatc1*[f/f] x dlck-cre mice. Above, immune blots showing the lack of NFATc1 expression in cytoplasmic (CP) and nuclear (N) proteins of aCD8[+] Ts from *Nfatc1*[f/f] x dlck-cre mice (lanes 3 and 4), and the expression of c.n.NFATc1/αA (point) in cells from *c.n.Nfatc1/αA x Nfatc1*[f/f] x dlck-cre mice (lanes 5 and 6). In lanes 1 and 2, proteins from WT cells were fractionated. Right, RT-PCRs showing *Il2* and *Myc* RNA levels in aCD8[+]Ts from WT, *Nfatc1*[f/f] x dlck-cre and *c.n.Nfatc1/αA x Nfatc1*[f/f] x dlck-cre mice. Data of 5 PCR assays are shown, relative to naïve WT CD8[+] T cells and normalized to *Actb*. Data are shown as means ± SEM. Unpaired Student's *t*-test was used for statistics. **g** Extracellular flux analysis of CD8[+] T cells from WT and *Nfatc1*[f/f] x CD4-cre mice stimulated by αCD3/CD28 (5/2 μg/ml) without (*above*) or with 100 U/ml hIL-2 (*below*) for 3 d. All assays were performed three times or more

within and around the *Il3*, *Ifng* and *Il2* genes, NFATc2 ablation resulted in very different effects on their expression. While NFATc2 ablation did not affect *Il2* induction, it reduced *Ifng* induction to approximately 55% and inhibited almost completely *Il3* induction in CTLs (Fig. 5e).

**NFATc1 ablation affects CD8$^+$T cell differentiation**. *Nfatc1$^{-/-}$* aCD8$^+$ T cells show a decrease in proliferation compared to WT and *Nfatc2$^{-/-}$* cells. Using a low concentration of plate-bound αCD3 (0.05 and 2 μg/ml αCD28) we detected a marked difference in proliferation between WT, *Nfatc1$^{-/-}$* and *Nfatc2$^{-/-}$* aCD8$^+$ T cells. In contrast, all three types of CTLs did not display any difference in proliferation in medium containing high (100 U/ml) or low concentrations of IL-2 (10 U/ml; Fig. 6a). Among the numerous genes encoding cell cycle regulators, only the *Cdkn1a* gene encoding the cell cycle inhibitor p21$^{WAW/CIP1}$ was found as NFATc1 target in aCD8$^+$ T cells. In aCD8$^+$ T cells, NFATc1 inactivation led to a decrease in *Cdkn1a* RNA levels to ~60%, and NFATc1/A bound within and upstream from the *Cdnk1a* gene (Fig. 6b, Supplementary Figs. 4c, d).

Several genes whose expression controls the differentiation to effector and memory CTLs are direct NFAT targets. T-bet and Eomesodermin (Eomes) are key transcription factors controlling CD8$^+$T cell differentiation[6]. In *Nfatc1$^{-/-}$* CTLs + the RNA level of the *Tbx21* gene encoding T-bet corresponds to approximately 30% of the WT level, and a lower T-bet protein expression was detected in *Nfatc1$^{-/-}$* (and *Nfatc2$^{-/-}$*) CTL + cells (Fig. 6c–e). NFATc1 (and NFATc2) binds to two intronic sites and strongly to one site 11 kb upstream of the *Tbx21* gene (Fig. 6f). In contrast, the *Eomes* gene is not bound by NFATc1, and its expression was found to be strongly enhanced in *Nfatc1$^{-/-}$* CTLs (Fig. 6c–e) reflecting the reciprocal expression of both transcription factors[23]. NFATc1 also binds to and controls in aCD8$^+$ T cells the *Il12rb1* and *Il12rb2* genes (Supplementary Fig. 5a) encoding the IL-12 receptor that enhances *Tbx21* expression via mTOR[24].

In *Nfatc1$^{-/-}$* aCD8$^+$ Ts only one third of the *Gzmb* transcripts were generated compared to WT aCD8$^+$ T cells, and NFATc1 binds to several sites of the *Gzmb* gene (Supplementary Fig. 5b). The transcript levels of *Prf1* and *Gzmb* genes encoding the most prominent perforins and granzymes in CD8$^+$ T cells increases 10-fold from aCD8$^+$ Ts to CTLs but NFATc1 ablation led unaffected their expression in CTLs (Fig. 6c).

Among 'fate markers' for memory T cells, the *Ccr7* and *Il7r* genes are NFATc1 targets. NFATc1 ablation led to a 4-5-fold decrease in *Il7ra* transcripts in CTL + cells encoding the IL-7 receptor α chain CD127, a marker for long-living memory T cells[25]. NFATc1 binds also to the *Itgae* gene encoding the integrin α$_E$/CD103 that is highly expressed on tissue-resident memory T cells, and NFATc1 ablation led to a decrease in *Itgae* transcript levels in CTLs (Figs. 6c, g; Supplementary Fig. 6). Furthermore, NFATc1 target genes are the *Irf4* and *Tcf7* genes that control CTL activity and differentiation[26, 27] (Fig. 6g; Supplementary Fig. 6).

Among the genes encoding cytoskeleton proteins the *Actb*, *Tuba1b*, *Tubb5* and *Vim* genes are highly transcribed in aCD8$^+$ T cells but only the level of *Tubb5* and *Vim* transcripts decreased to 61.5 and 40% in *Nfatc1$^{-/-}$* aCD8$^+$ Ts. In *Nfatc1$^{-/-}$* CTLs, the transcript levels of *Actn1* and *Plek* genes were strongly impaired, compared to WT cells, and both genes are bound by NFATc1 (Supplementary Fig. 7a, b).

**NFATc1 controls the metabolism of CD8$^+$ T cells**. The transcript level of almost all genes controlling glycolysis[28] is significantly lower in *Nfatc1$^{-/-}$* than WT or *Nfatc2$^{-/-}$* aCD8$^+$ T cells (Fig. 7a; Supplementary Fig. 8). This observation

prompted us to study the role of NFATc1 in metabolism regulation of CD8$^+$ T cells. Due to their high proliferation rate, activated T cells have a high demand for energy and cellular building blocks, as nucleotides and amino acids and, therefore, switch their metabolism from OXPHOS to glycolysis[29]. In RT-PCR assays we determined less than 30–35% of WT transcript level for the *Slc2a1*, *Slc2a3* and *Hk2* genes encoding the glucose transporters Glut1 and Glut3, respectively, and hexokinase 2 (Hk2), the enzyme responsible for the fixation of glucose in cells (Figs. 7a, b; Supplementary Fig. 8). Using the fluorescent glucose analog 2-[N-(7-nitrobenz-2-oxa-1,3-diazol-4-yl) amino]-2-deoxy-D-glucose (2-NBDG) we observed a 70% reduction in glucose uptake into *Nfatc1$^{-/-}$* aCD8$^+$ Ts, compared to WT, whereas only a slight decrease was detected for *Nfatc2$^{-/-}$* cells (Fig. 7c). The decrease in glucose incorporation was accompanied by a decrease in extracellular acidification rate (ECAR) of *Nfatc1$^{-/-}$* aCD8$^+$ Ts, as measured by extracellular flux analysis. For aCD8$^+$ Ts generated with a low concentration of αCD3 (0.05 and 2 μg/ml αCD28) a 5-fold increase in glycolysis and glycolytic capacity was observed, while cells stimulated with standard concentrations (5 μg/ml αCD3, 2 μg/ml αCD28) displayed a three fold increase in both parameters. However, *Nfatc1$^{-/-}$* aCD8$^+$ Ts stimulated with either low or high concentrations of αCD3/CD28 exhibited lower glycolysis and glycolytic capacity of 40–60% of WT level (Fig. 7d).

Contrary to the effect on glycolysis, NFATc1 ablation exerted only a moderate effect on mitochondrial respiration. In mito stress tests, the oxygen consumption rate (OCR) differed only slightly between WT and *Nfatc1$^{-/-}$* aCD8$^+$ Ts, although *Nfatc1$^{-/-}$* aCD8$^+$ Ts showed a tendency towards increased maximal respiration, compared to WT cells (Fig. 7e), and the RNA levels of genes encoding mitochondrial OXPHOS enzymes were slightly enhanced (Supplementary Fig. 9a).

In part, the decrease in glucose incorporation of *Nfatc1$^{-/-}$* aCD8$^+$ Ts could be restored by overexpressing a constitutive nuclear (c.n.) version of NFATc1/αA. For this purpose we analyzed aCD8$^+$ T cells from *c.n.Nfatc1/αA x Nfatc1$^{f/f}$ x* dlck-cre mice. Due to cre expression directed by the distal *lck* promoter[30] these mice express c.n.NFATc1/αA upon removal of a 'floxed' STOP sequence from the Rosa26 locus[31] in the absence of endogenous NFATc1. Those cells revealed a moderate, but reproducible increase in their glucose uptake, and in *Il2* and *Myc* RNA levels (Fig. 7f).

The effect of NFATc1 ablation on glycolysis led to the question of whether NFATc1 exerts a direct effect on glycolytic genes. However, in ChIP seq assays apart from the *Hk2* and *Gapdh* genes (Supplementary Fig. 10a) no further genes of the glycolysis cascade showed distinct NFATc1 binding. One mediator of the NFATc1 effect could be IL-2 whose expression is strongly diminished in *Nfatc1$^{-/-}$* T cells (Fig. 5e). IL-2 is known to activate multiple metabolic and transcriptional programs in T cells, mainly through the Ser/Thr kinase mTORC1, a prominent IL-2 target in T cells[32]. When we tested the effect of IL-2 in splenic CD8$^+$ T cells on 4E-BP1, a direct target of mTOR signals, we observed a marked increase in TCR-mediated phosphorylation that was inhibited by rapamycin. However, NFATc1 ablation was without effect on 4E-BP1 phosphorylation (Supplementary Fig. 9b). Nevertheless, adding 100 U/ml IL-2 to CD8$^+$ T cell cultures that were activated by αCD3/CD28 for 3 d revealed a marked increase in glycolysis of NFATc1-deficient cells (Fig. 7g). This finding is supported by similar RNA levels of glycolytic genes in WT and *Nfatc1$^{-/-}$* CTLs that were cultured in IL-2-containing medium for 6 d. Although we observed a drop in *Slc2a1* and *Hk2* RNA levels, we observed a similar glucose uptake and glycolysis of these cells (Supplementary Fig. 9c–f). Even upon re-stimulation of CTL- cells by αCD3/CD28 for 16 h, no

differences in glycolysis and glycolytic capacity were detected between both types of cells (Supplementary Fig. 9f).

These findings do not exclude that NFATc1 affects also the metabolism of CTLs. The strong NFATc1 binding to the *Slc7a5* gene (Supplementary Fig. 10a) encoding a leucine transporter that coordinates the metabolic reprogramming of CTLs[33] suggests that in CTLs NFATc1 controls metabolic pathways other than glycolysis.

While IL-2 enhanced the level of glycolysis of *Nfatc1*[−/−] aCD8[+] Ts to that of *Nfatc2*[−/−] cells, it did not completely rescue glycolysis to the level of WT cells. This suggests that genes in addition to *Il2* that affect glycolysis might be controlled by NFATs. Candidates are the *Irf4* and *Myc* genes. Both control the metabolic re-programming of T cells upon TCR stimulation[26, 28] and their RNA levels were markedly diminished in *Nfatc1*[−/−] aCD8[+] Ts, and the *Irf4* gene showed NFATc1 binding (Fig. 6g; Supplementary Figs 6 and 10b).

## Discussion

The results of our study show that NFATc1 controls the effector function of CD8[+] T cells at multiple levels. *Nfatc1*[−/−] aCD8[+] T cells exhibit a decrease in proliferation and metabolic switch from OXPHOS to glycolysis, and—at the molecular level—in a poor RNA expression of genes coding for numerous glycolytic genes. This is mainly due to the missing IL-2 whose expression is induced by NFATc1. The diminished expression of *Tbx21*, encoding a key transcription factor of effector CD8[+] Ts, and of *Gzmb* gene are further molecular signs of defective function of *Nfatc1*[−/−] CD8[+] T cells. Upon activation, *Nfatc1*[−/−] CTLs exhibit an impaired depletion of apical F-actin and recruitment of lytic granules and mitochondria to the IS, and in a strong reduction in induction of numerous cytokine and chemokine genes. These include, apart from the *Il2* gene, the genes encoding IFN-γ and IL-3, and the chemokine *Ccl3*, *Ccl4*, and *Xcl1* genes (Fig. 4). These defects culminate in the defective eradication of infected and malignant cells.

In CTLs, NFATc1 and NFATc2 overlap in their binding to almost all genes (and sites) but differ strongly in their transcriptional activity (Figs 4 and 5). These data seem to contradict published data on the role of NFATs, in particular of NFATc2 (NFAT1), in the exhaustion of CD8[+] T cells[19], an anergy-like state of activated CD8[+] T cells chronically infected with viruses. However, they are not necessarily in conflict with a promoting role of NFATs in CD8[+] T cell exhaustion. NFATc1 and NFATc2 differ in their expression in activated lymphocytes. While short immune receptor signals trigger the rapid translocation of both factors into the nucleus, more persistent signals induce the transcription of *Nfatc1* gene leading to the massive synthesis of NFATc1/αA within a few hours (Fig. 3d). Although within several days the levels of NFATc2 also increase by persistent TCR signals (Fig. 3e), signals other than TCR triggering result in the further increase of NFATc2 in peripheral T cells[34]. These and further data led us to conclude that the *Nfatc1* gene codes for (at least) two factors which differ remarkably in their transcriptional properties[35]. In our view (i) NFATc1/αA which is highly expressed in activated T cells (Fig. 3d) supports 'immunity', i.e. the effector function of lymphocytes, including glycolysis, whereas (ii) NFATc1/ßC (and other longer NFATc1 isoforms) which is mainly expressed in resting T cells, supports the induction of 'tolerance' and 'exhaustion' of lymphocytes[35]. The latter shares peculiarities with NFATc2, such as two SUMOylation sites within the C-terminal domain that is missing in NFATc1/αA, and SUMOylation converts NFATc1/ßC to a suppressor of IL-2 induction[36].

In a study on the ablation of NFATc1 (NFAT2) in murine CD8[+] T cells, Pachulec et al.[11] described an increase in the percentage of CD8[+]CD44[high]CD122[+] cells in thymus, spleen and LNs of unchallenged *Nfatc1*[f/f] x CD4-cre mice which the authors classified as innate-like CD8[+] T cells with characteristics of conventional memory CD8[+] T cells. In our Listeria infection assays, we observed a slight increase in the number of *Nfatc1*[−/−] T[CM], compared to WT cells, and a decrease of *Nfatc1*[−/−] T[EM] cells (Fig. 2e). The latter showed a decrease in their cytotoxicity (Supplementary Fig. 3a), similar to the findings published by Pachulec et al.[11] for *Nfatc1*[−/−] CD8[+]T cells.

We focused our attention on the molecular events controlled by NFATc1 (and NFATc2) during the activation and differentiation of CD8[+] T cells to CTLs in vitro generated mostly from *Nfatc1*[f/f] x CD4-cre mice as well. Those mice showed a normal development of peripheral CD4[+]/CD8[+] T cells[37]. Several of our data were also confirmed by using *Nfatc1*[f/f] x dlck-cre mice in parallel, in which the *Nfatc1* gene was inactivated in peripheral T cells, i.e. after thymic selection[30]. Taken together, our NGS and ChIP seq data on the genome-wide gene control by NFATc1 in CD8[+] T cells confirm and extend the data by Pachulec et al.[11] on the NFATc1-mediated control of IFNγ production in CD8[+] T cells.

NFATc1 binds to numerous genes controlling the activity and fate of CD8[+] T cells, but several physiological events of CD8[+] T cells are also indirectly controlled by NFATc1. One example is the metabolic switch from OXPHOS to glycolysis during T cell activation, others might be the diminished proliferation (Fig. 6a) and enhanced apoptosis of *Nfatc1*[−/−] lymphocytes[38]. In *Nfatc1*[−/−] CD8[+] T cells, the expression of numerous genes encoding glycolytic enzymes are impaired, and glycolysis is decreased to less than 50% of normal level in WT cells. Adding IL-2 to aCD3/CD28-stimulated *Nfatc1*[−/−] CD8[+] T cells restored the defect to a large extent. In line with published studies[39–41] this implies that in addition to TCR signals IL-2 supports the metabolic switch from OXPHOS to glycolysis during primary T cell activation.

In ChIP seq assays both NFATc1 and NFATc2 were found to bind at and near the promoter of the *Il2* gene (Fig. 5e). However, NFATc2 ablation did not-or very mildly-affect *Il2* expression, whereas NFATc1 ablation strongly impaired *Il2* induction in both aCD8[+] Ts and CTLs + . It is currently unclear why NFATc1—and not NFATc2—ablation exerts such a deleterious effect on *Il2* transcription. One explanation might be the high concentrations of NFATc1 in aCD8[+] T and CTL + cells upon TCR stimulation (Fig. 3d). It is known that the *Il2* gene needs a certain threshold level for NFATs to be transcribed[42, 43] which might be reached by NFATc1, but not NFATc2 expression. In contrast to this all-or-none expression of the *Il2* gene, the transcription of *Ifng* and *Il3* genes seems to follow a gradual way that is affected by either NFATc1 or NFATc2 ablation (Fig. 5e).

In addition to IL-2, the diminished expression of other factors, such as of IRF4 and c-Myc, might also contribute to the impaired metabolic switch to glycolysis in *Nfatc1*[−/−] aCD8[+] Ts. The TCR-mediated expression of c-Myc is enhanced by IL-2 signals[44], and—similar to Irf4—c-Myc was shown to bind to and to enhance the expression of numerous glycolytic genes[28]. A further candidate is the NFAT target gene *Pdcd1* encoding the repressor PD-1 that was shown to affect glycolysis[45]. In a recent study high concentrations of intracellular L-arginine were reported to suppress glycolysis and to generate central memory-like T cells[46].

Numerous cytokine and chemokine genes are direct NFATc1 targets, and by regulating those genes NFATc1 seems to control the mobility and motility of CTLs and, thereby, their cytotoxicity. It is unknown whether such NFATc1-dependent key molecules control also the cytoskeleton re-organization and recruitment of

mitochondria and lytic granules to the IS. The interplay between mitochondria and Orai1 channels at the IS is highly relevant for the generation of local (at the IS) and global $Ca^{2+}$ signals (in the cytosol and in the nucleus). Mitochondria take up $Ca^{2+}$ and reduce local $Ca^{2+}$ domains at the IS, which in turn inhibit $Ca^{2+}$ dependent inactivation of Orai1 channels[47, 48]. Thus, mitochondria control local and global $Ca^{2+}$ signals following TCR induced $Ca^{2+}$ entry by Orai1 channels. By controlling $Ca^{2+}$ signals, mitochondria regulate NFAT translocation into the nucleus in $CD4^+$ lymphocytes[48].

While we did not detect defects in the expression of numerous formin genes that orchestrate the formation of cytoskeleton[49], in CTLs a strong reduction was observed in the expression of NFATc1-target genes *Actn1* and *Plek* (Supplementary Fig. 6). The expression of *Actn1* gene encoding α-actinin 1 contributes to T cell activation by forming multiprotein complexes with Lck and Rack1 that couple actin skeleton with TCR triggering, and it was shown to participate in T cell migration[50, 51].

Taken together, our study revealed numerous novel NFATc1 targets in cytotoxic T cells. The detailed knowledge of their NFAT-mediated control will help to interfere with the molecular mechanisms that control the activity of cytotoxic T cells to eradicate infected and malignant cells.

## Methods

**Mice and T cell activation**. All mice used in the experiments were 8-12 weeks old, at C57/B6 background and sex- and age-matched. The T cell-specific CD4-cre mice have been described[52]. Two *Nfatc1^{f/f}* mouse lines (also designated as *Nfat2^{f/f}* mice[19, 37, 53]) and conventional *Nfatc2^{−/−}* mice[54] were used. The *c.n.Nfatc1/αA x Nfatc1^{f/f} x* dlck-cre mice express a constitutively active version of NFATc1/αA from the Rosa26 locus upon removal of a 'floxed' STOP sequence[31] in peripheral T cells. Animal experiments were performed according to project licenses (55.2–2531.01-53/10B and -76/14), which are approved by the Regierung von Unterfranken, Würzburg. For $CD8^+$ T cell isolation, the mouse CD8 (Ly2) microbeads kit #130-049-401 kit (Miltenyi Biotech) was used according to the manufacturer's protocol. After purification, the purity of $CD8^+$ T cells was checked by flow cytometry. Cells were suspended in X-vivo medium to a final concentration of $2 \times 10^6$/ml. For αCD3/CD28 stimulation, if not stated otherwise 5 µg CD3ε (clone 37.51) and 2 µg CD28 (clone 145-2C11) (both BD Pharmingen) were diluted in 1 ml PBS which was used to coat multi well plates. T cells were also stimulated with 10 ng/ml TPA and 0.5 µM ionomycin (normally for 5 h). To generate CTLs, $CD8^+$ T cells were either treated by αCD3/CD28 for (2-) 3 d followed by incubation for (5-) 6 d in medium containing 100 U/ml hIL-2. In some assays, $CD8^+$ T cells were co-cultured for 6 d with splenocytes from a Balb/c mouse that were irradiated (by 30 Gy, 5′, 16 kV).

**Cytotoxicity assays**. In the first assay (Fig. 2a), the mineral oil induced murine MOPC-315.BMP2.FUGLW[55] plasmacytoma cell line (H2$^b$) expressing eGFP and luciferase was used as target cell line for killing assays. This cell line originates from an allogeneic donor and, therefore, allo-reactive B6 CTLs recognizing the MHC mismatch induce target cell death. To measure luciferase during co-culture with B6 $CD8^+$ T cells, cells were re-suspended and aliquots were transferred into 1.5 ml Eppendorf tubes. After centrifugation, the cell pellet was washed once with PBS and finally resuspended in 100 µl harvesting buffer to lyse cell membranes. After centrifugation, 50 µl of supernatant was transferred into a white, non-transparent 96-well plate. The LUMIstar Omega was primed with 1 ml of ready-to-use luciferin solution before luciferase-activity measurement was performed. 50 µl of luciferin solution per well were automatically added to the sample and measurement was performed.

In the second assay (Fig. 2b) $10^4$ cells A20J cells labeled with CFSE were cultivated in 100 µl X-vivo medium with $10^5$ CTLs at 37° for 4 h, followed by staining with Abs for CD8a-biotin and Streptavidin-APC and flow cytometry. Prior analysis, cells were also stained by PI to distinguish living A20J target cells (CFSE$^+$, PI$^−$) from dead cells (CFSE$^+$, PI$^+$).

**Generation of Nfatc1/A-Bio BAC tg mice**. *Nfatc1/A-Bio* BAC constructs were generated by insertion of a C-terminal tag sequence for in vivo biotinylation[56] into the BAC RP23-361H16 containing the murine *Nfatc1* gene (mm9 chr.18, 80,779,051–80,993,617, 214 kb) in front of the stop codon of exon 9. In a first step a targeting vector was generated with 5′ (1 kb) and 3′ (1 kb) homology regions flanking the tag sequence. The homology arms were generated by PCR using the primers E9BoxA_AscI1kb_ubiofwd: 5′-TAGGCGCGCCTCTGAAATCCCCAGC AGAAAT-3′, E9BoxA_Ex9-bio_transrev: 5′-TTGGTAAAAACCTCCTCTCAG CTC-3′ for the 5′ and E9BoxB_bio-down_transfwd: 5′-TCAAACGCCGGAGGCT

CGTGAGCAGCCCCCCGAGGCTATAAG-3′, E9BoxB_NotI1kb_dbiorev: 5′-TAG CGGCCGCCTGATAAGACAATTGGTTTCA-3′ for the 3′ region. The center tag sequence was amplified with the primer pair Ex9bio_fwd: 5′-gagctgagaggagggtttt tacCAATTGGGCGGTGGAGGTCTG-3′, E9BoxB_NotI1kb_dbiorev: 5′-TAGCG GCCGCCTGATAAGACAATTGGTTTCA-3′. In a combined PCR using the amplified fragments a fusion product was generated and cloned as an AscI and NotI fragment into the shuttle vector pLD53.RecA[57]. After RecA mediated recombination BAC DNA was tested for insertion of the tag sequence by two PCR reactions with the primer pairs Ex9BoxAupst: 5′-ACCACACGAGGAAAGAA ACG-3′, Ex9BioRev.: 5′-TCTTCTGCGAGTCGAGGATT-3′ (1235 bp) and Ex9BioFwd: AATCCTCGACTCGCAGAAGA, Ex9BoxBdwn: 5′-ACAGACAGTT GCTGCCCTTT-3′ (1143 bp) and by sequencing. NruI linearized BAC DNA was microinjected into blastocysts. Mouse biopsies were tested by PCR for BAC integration with the primers geEx9-bio-for: 5′-GTTTTTACCAATTGGGCGGT GGA-3′, geEx9-bio-rev: 5′-CTCCATCTTCTGCGAGTCGAGGA-3′ (209 bp). The 5′-end of integrated BAC DNA was detected with the primer pair 5′BAC(T7)for: 5′-GGTCCATCCTTTTGTCTCA-3′ and 3′e3.6(T7)rev: 5′-CGAGCTTGACATT GTAGGA-3′ (512 bp) and the 3′-end with the primers 5′e3.6(SP6)for: 5′-CGT CGACATTTAGGTGACA-3′ and 3′BAC (SP6)rev: 5′-CCATCGTTCCCTGAC TCA-3′ (439 bp).

**Bio-ChIP analysis of NFATc1/A binding**. $CD8^+$ T cells were isolated from spleen of *Nfatc1/A-Bio^{+/−} x Rosa26BirA^{+/−}* or *Rosa26BirA^{+/−}* mice. After a primary stimulation with plate-bound αCD3 (5 ng/ml) and αCD28 (10 ng/ml) for 48 h, cells were harvested and cultivated for additional 3 d with medium containing 100 ng/ml hIL-2. Re-stimulation was performed by addition of TPA (5 ng/ml) and Ionomycin (0,5 µM) for 5 h. Chromatin from $1 \times 10^7$ fixed CTLs was prepared using a lysis buffer containing 1% SDS and sonicated by a Vibra-Cell VCX 130 (Sonics, CT, USA) for 16′ before precipitation of chromatin by streptavidin beads (M-280, #11205D, Thermo Fisher Scientific, MA, USA), essentially as described[58]. The precipitated genomic DNA was controlled for fragment size by the Agilent Bioanalyzer using a High Sensitivity DNA Chip according to the manufacturer's instruction (Agilent Technologies, CA, USA) and quantified by a Qubit 2.0 fluorometer using a ds DNA HS assay kit (Thermo Fisher Scientific, USA, Q32851). 7 ng of precipitated genomic DNA from *Nfatc1/A-Bio^{+/−}* and *Rosa26BirA^{+/−}* cells was used for library preparation (NEBNext Ultra DNA Library Prep Kit for Illumina, NEB) according to manufacturer's instructions. The quantity of final library was assessed by Qubit 2.0 fluorometer using a dsDNA HS assay kit, and the average size was determined by Agilent's Bioanalyzer on a high sensitivity DNA Chip. Sequencing of the final libraries was carried out as a 50 bp single read run on an Illumina HiSeqTM2500 (Illumina, San Diego, USA) by the Institute of Molecular Genetics (JGU, Mainz, Germany) using a TruSeq Rapid SBS Kit v2 and a HiSeq Rapid Flow Cell v2. RTAVersion 1.18.64 was employed for cluster identification. Analysis was performed on usegalaxy.org[59]. 50 bp sequence reads that passed the Illumina quality filtering were aligned to the mouse genome assembly version of July 2007 (NCB I37/mm9), using the Map with Bowtie for Illumina 1.1.2. Results were visualized with the "Integrative Genomics Viewer" IGV version 2.3.81[60]. We used the MACS program (Galaxy Version 1.0.1)[61] with default parameters, a genome size of 2,700,000,000 bp (mm9) and the reads from Rosa26BirA+/− IP as control sample to identify peaks that were filtered for P values of $< 10^{−5}$. A P-value of $< 10^{−12}$ was used for comparison to NFATc2 binding[19]. Motif discovery was performed with "Discriminative Regular Expression Motif Elicitation" (DREME) version 4.11.2[62]. Genes nearby 100 kb to annotated peaks were identified with GREAT version 3.0.0[63].

**Infections with Listeria monocytogenes**. For *Listeria* infections in vivo, an attenuated Listeria strain *Lm-Ova ΔActA* was used. The strain is defective in the expression of the ActA protein that mediates normally direct cell-to-cell transmission of replicated bacteria, and thereby increases virulence up to 1000 fold[25]. $5 \times 10^5$ colony forming units (CFU) from a cryo-conserved vial stock were suspended in 200 µl PBS and injected i.p. into either *WT* or *Nfatc1^{f/f} x* CD4-cre mice. Mice were monitored daily, until they were sacrificed at d 5 post-infection. Livers were gently meshed in PBS containing 0.1% TritonX-100 to destroy cellular membranes. Serial 1:10 dilutions were made in PBS and 100 µl each were plated out on agar plates selective for Listeria, and incubated at 37 °C for 2 d.

**Immunoblotting and PCR assays**. Immune blots were performed with whole protein extracts on PAGE-SDS gels followed by detection of NFATc1 using 7A6 mAb #556602, or for NFATc2 mAb #5062574 (both BD Pharmingen). For detecting NFATc1/α, a pAb raised against the NFATc1α-peptide Ab #IG-457 (ImmunoGlobe) was used. As loading control, filters were stained by Ponceau Red (PR in Fig. 7f) and/or re-probed with the mAb #ab8227 specific for β-actin (Abcam). Signals were developed using a chemiluminescence detection system (Thermo Fisher Scientific).

For RT-PCR assays, RNA was isolated from washed and deep-frozen lymphocytes using a standard TRIzol/isopropanol protocol. cDNAs were synthesized using the iScript cDNA synthesis kit according to the manufacturer's instructions (Bio-Rad). Real-time PCR assays were performed using the SYBR

green master mix (Applied Biosystem) using the primers presented in Supplementary Table 1, and as described previously[64].

**Flow cytometry**. T cells were washed once in cold PBS containing 0.1% BSA (FACS buffer) before blocking with anti-FcγRII/FcγRIII (2.4G2, BD Pharmingen, San Diego, CA). Stainings were performed on ice using conjugated mAbs (eBioscience, San Diego, CA, if not stated otherwise), diluted 1:300 (1:200 for intracellular cytokines) in FACS buffer followed by incubation for 20 min. After washing with FACS buffer, cells were analyzed on a FACS Canto II (BD) and FlowJo software (Tree star, Ashland, OR). The following Abs were used: CD8-FITC (#11-0081-82), IL-2-APC (#17-7021-81), IFNγ-APC (#17-7311-82), CD62L-PE (#12-0621-81), CD44-APC (#17-0441-81), Perforin-APC (#17-9392-80), Granzyme B-PE (#12-8898-80). Abs against TNF-PE (#130-092-245), T-bet-PE (#130-098-653) and IL-17-PE (#130-094-296) were from Miltenyi Biotec. For intracellular staining, the fixation and permeabilization kit (Plus Brefeldin A; eBioscience, Cat. no. 88-8823-88) was used according to manufacturer's recommendation.

**RNA seq assays**. RNA from deep-frozen CD8[+] T cells was extracted using Trizol. Quantity of total RNA was assessed with the Qubit 2.0 and quality was checked using a RNA 6000 Nano chip on Agilent's bioanalyzer. Sequencing libraries were prepared from 600 ng total RNA using a NEBnext ultra RNA library prep kit following the manufacturer's instruction. The resulting barcoded cDNA libraries were sequenced on an Illumina HiSeq 2500 platform for 50 nucleotides (single end).

**Imaging of cytoskeleton dynamics in CTLs**. CTLs in warm RPMI/0.5% BSA medium were activated on glass bottom chamber slides (μ-slides, Ibidi, Germany) coated with αCD3/CD28 for the indicated time points. Co-stimulation was immediately stopped by covering cells with a pre-warmed fixation solution of 4% formaldehyde and 0.05% glutaraldehyde in PBS for 10 min at 37°. Samples were rinsed in PBS, permeabilized with 0.1% Triton X-100 for 5 min, treated with 50 mM NH$_4$Cl for 10 min and blocked with 5% BSA. For visualization of filamentous F-actin cells were stained with phalloidin 488 (Actin-skeleton, Cytoskeleton) and for MTOC reorientation with an anti-β-tubulin primary Ab (clone 9F3, Cell Signaling) overnight at 4° followed by incubation with anti-rabbit-Alexa 568 secondary Ab (Invitrogen) for 45 min at RT and a final step with DAPI for nuclear staining. Preparations were stored in PBS for total reflection (TIRF) microscopy and data were obtained by using a Leica AM TIRF system with software AF 6000 LX, equipped with a 100x oil objective (HCX PL APO, NA 1.47) and laser lines 488 and 561 nm. For confocal laser scanning microscopy (CLSM) samples were covered with mounting medium (Ibidi). Images were acquired on a Zeiss LSM 780, provided with 40x and 63x oil objectives (Plan-Apochromat, NA 1.4) and laser excitation at 405, 488 and 561 nm and processed with software ZEN 2012.

To quantify the size of F-actin depletion zone imaged by TIRF microscopy and the increase in cellular spreading taken by CLSM on Ab-coated surfaces, cell area was calculated after setting a manual threshold and measured in square micrometer using the software platform Fiji[65]. For morphological examination under a field scanning electron microscope (SEM), CTLs were co-stimulated using 12 mm glass slices in a 24-well plate, fixed in pre-warmed 6.25% glutaraldehyde in 50 mM phosphate buffer (pH 7.2) for 15 min at RT and subsequently overnight at 4 °C. After three washing steps in PBS, samples were prepared by stepwise dehydration in acetone, then critical point dried, sputtered with gold/palladium and imaged on a JSM 7500 F (Jeol).

**Imaging of recruitment of lytic granules and mitochondria**. C57/B6 CD8[+] T cells were purified and stimulated with irradiated BALB/c splenocytes as described above. On d 5 after isolation, CTLs were sorted on a BD FACSAria TMIII (70 μm nozzle, gating on FSC and SSC, cooled reservoir) to remove dead cells and debris. CTLs were cultured overnight in X-vivo medium, loaded with either 400 nM LysoTracker Green DND-26 and 100 nM MitoTracker Deep Red FM or 400 nM Lysotracker Red DND-99 and 100 nM MitoTracker Green FM for 30 min at 37 °C, washed and transferred into a black clear-bottom imaging microplate (Greiner 781096, 384 wells) in X-vivo medium containing either 400 nM LysoTracker Green DND-26 or 400nM LysoTracker Red DND-99 (1.3 × 10$^5$ CTL/well). Shortly before the start, 2 × 10$^4$ Dynabeads (Mouse T-Activator CD3/CD28 beads; Life Technologies, 11456D) were added to each well. Time-lapse imaging was performed on a Zeiss AxioObserver at 37 °C and 5% CO$_2$ with a Fluar 20 × /0.75 objective, using transmitted light and epifluorescence with filter set 38 HE (Ex 470/40, Em 525/50) and a custom filter set (Ex 620/60, Em 700/75). Background fluorescence was subtracted, and the polarization of mitochondria and lysosomes was quantified (assuming stable polarization when the fluorescence signal was close to the contact zone for more than 3 images (or 60 s)). For the intensity profiles shown in Supplementary Fig. 1c, cells in stable contact with beads for at least 10 frames (200 s) were analyzed if cell location was plane to the contacting bead (the cell did not lie above or below the bead), no labeled organelles were located above or below the bead and the cell diameter was larger than 5 μm in order to faithfully localize the proximal and distal part of the cell. The selected cells

at the chosen time points were analyzed with Fiji ImageJ 2.0 using the "Line Tool" (for round or elongated cell morphology) or the "Segmented Line Tool" (for cells with non-symmetric morphology). A line was drawn from the contacting bead towards the center of the IS and further along the midline of the cell body to the rear of the cell and intensity values were measured along this line.

**Metabolic assays**. In extracellular flux assays, ECAR of CD8[+] T cells was measured with a XF96 analyzer (Seahorse Bioscience). 4 × 10$^5$ freshly prepared CD8[+] T cells from WT, Nfatc1$^{f/f}$ x CD4-cre and Nfatc2$^{-/-}$ mice were maintained overnight without stimulation or stimulated by αCD3/CD28. In parallel, the same number of CTL- cells were stimulated for a further day by αCD3/CD8. Upon counting, the cells were seeded in a plate pre-coated with poly-D-lysine (Sigma; 50 μg/ml) in XF Seahorse medium with glutamine alone. Then, 10 mM glucose was injected, followed by 1 μM oligomycin and 20 mM 2-deoxyglucose. Mito stress test assays for the determination of oxygen consumption rate (OCR) were performed in a similar way according the manufacturers protocol (Seahorse Bioscience). 2-NBDG (Cayman Chemical) assays were performed by incubation of cells with 50 μM 2-NBDG for 1 h at 37 °C followed by flow cytometry.

**Statistical analysis**. Statistical analyses were performed using GraphPad (Prism) software, version 6.0. Data presented as mean and error bars in figures represent ± SEM. Unpaired t-tests were performed to evaluate the statistical significance of the data set. Statistical significances were calculated and indicated (***$p < 0.001$, **$p < 0.005$ and *$p < 0.05$).

**Data availability**. Sequence data that support this study have been deposited in NCBI's Gene Expression Omnibus and are accessible through GEO Series accession number GSE98726. For analysis of NFATc2 binding, data from GSE64409 were used. All other data supporting the findings of this study are available with the article, and can also be obtained from the authors.

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

## Acknowledgements

We are very much indebted to Doris Michel for technical support. We are particularly thankful to Dr Meinrad Busslinger (Vienna) for materials and advice regarding the Bio-tag technology, and to Dr Anjana Rao (San Diego) for providing us with *Nfat2/Nfatc1$^{f/f}$* mice before publication. We are indebted to Drs Antje Gohla and Georg Krohne (both Würzburg) for support in TIRF and scanning electron microscopy, respectively. For further mouse lines and reagents, we wish to thank Drs V. Ellenrieder (Göttingen) and L. Glimcher (New York). We thank Dr A. Rosenwald for his kind support. We also thank Dr E. Krause (FACS Facility, CIPMM, Homburg) for help with cell sorting.

The work was supported by the Deutsche Forschungsgemeinschaft, grant TRR52 (to E.S. and A.A.), grant SFB 894, project A1 (to M.H.), the Wilhelm-Sander foundation (to E.S. and S.K.-H.), the Scheel foundation for Cancer Research (to E.S. and A.A.) and the IZKF Würzburg (to A.A.).

## Author contributions

S.K.-H., K.M., M.K., T.P., R.R., J.F., M.Q., M.V., C.K., C.B., R.S., U.H., A.P. and N.M. performed experiments, S.K.-H., K.M. A.B., M.H., T.B., F.B.-S., A.A. and A.Sch. designed experiments, analyzed data and supported the preparation of the manuscript, E.S. led the investigation and wrote the manuscript.

## Additional information

**Competing statement:** The authors declare no competing financial interests.

