## [Peer Review File · Nature Communications]

Reviewers' comments:

Reviewer #1 (Remarks to the Author):

This is a nice comprehensive study of the role of Nfatc1 on CD8 T cell activation and function. The inclusion of Nfatc2^{-/-} mice enhances the robustness of these findings. Furthermore, the authors link Nfatc induced immune activation with metabolic reprogramming. There are a couple of issues that need clarification:

1. While the finding is interesting the exact significance and mechanism surrounding the recruitment of mitochondria and cytotoxic granules to the IS in Figure 1E is unclear
2. Again the significance and mechanism of the generation of the different sizes of Nfat1c in Figure 3 is unclear
3. Throughout the paper the authors keep referring to genes controlling aerobic glycolysis this is wrong. These are simply genes involved in glycolysis.
4. Many of the findings can be controlled by mTOR activity. Have the authors examined mTOR activity in the knockout cells? Also, have they examined the role of Nfatc1 on Slc7a5 which is a calcineurin dependent leucine transporter?
5. I disagree with the comments in the second paragraph of the discussion regarding energy. If the authors insist on this view they must defend it better. Also in the discussion they overstate the role of their findings on metabolism. This should be tempered by the fact that there are multiple inputs to metabolic reprogramming. This includes the concept of metabolic syndrome for CD8⁺ T cells which I think is not developed well enough to include in this work.

Reviewer #2 (Remarks to the Author):

Nat Comms 16-28997

This manuscript addresses the interesting question of the role of NFATc1 in CD8⁺ cytotoxicity using a series of genetic models to examine the role of this NFAT transcription factor. A broad range of assays and technical approaches are used to examine the contribution of NFATc1 including imaging, functional (in vitro and in vivo) assays, transcriptional and protein interactions to provide support for a key role of NFATc1 in CTL effector function. While this is impressive the paper is difficult to review at the moment as more clarity as to which genetic models are used for which experiments.

The methods section describes two genetic models for NFATc1 deletion, using a floxed NFATc1 mouse which is stated to be described in citation 46 (Lee et al "A critical role for Dnmt1 and DNA methylation...") This does not seem to describe the generation of a floxed NFATc1 mouse, did they perhaps mean reference 47? What is really needed is the primary citation describing the exact construct for NFATc1 locus floxing (which neither citation provide). In the acknowledgements the authors mention that Anjana Rao has provided floxed mice "before publication." It is critical to know exactly what is being deleted for interpretation of the data. Without this information this paper cannot be properly reviewed.

The next key issue is which Cre was used to generate the mice used in each experiment and how efficient was the excision in each case? Was any protein produced in any of the cells used? Although

the investigators have antibodies to different splice products from NFATc1 there is almost no data to show loss of the protein.

It is important to know which Cre was used as the CD4 Cre will be activated during the double-positive stage of thymocyte development and may therefore cause defects in the development of mature CD8/CTL. The dLck-Cre comes on after positive selection of thymocytes and is therefore more appropriate for studying the role of NFATc1 in CTL. However, the efficiency of excision has been found to vary according to the floxed gene and it is vital to show efficiency of excision. Furthermore, use of this Cre is only described for Figure 7. Does this mean that all other experiments used the CD4-Cre? All of these issues need to be resolved before the paper can be properly assessed.

One other aspect that the authors may wish to consider is that at the moment the manuscript comes across as a collection of studies from a number of investigators that have been strung together in which the quality of the data varies. For example in Figure 1 the imaging in a-d is excellent, but the resolution in e is so greatly reduced that the statements regarding polarisation of lytic granules and mitochondria are not supported. Mitotracker and lysotracker should image discrete puncta within the cytoplasm rather than the diffuse signals shown.

Reviewer #3 (Remarks to the Author):

SUMMARY:

The authors studied the role of NFATc1 in controlling effector functions of CD8+T cells. In particular, they analyzed in vitro differentiated CD8+ cells and compared NFATc1^{-/-} and WT cells concerning cytotoxic effects, cytokine production, expression of NFATs, global gene expression, proliferation, and metabolic switch behavior.

Their main conclusion (title and summary) is that the transcription factor NFATc1 controls the cytotoxicity of murine CTLs. However, they neither fully characterized the cultured CD8 cells for composition in each experiment nor did they experiment with isolated defined Teff, Tem or Tcm cell subpopulations. This is a crucial point because each CD8 subpopulation has a different strength of cytotoxic capability and other effector functions (e.g. Fig 1. of Wherry and Ahmed, 2004). Therefore, differences of NFATc1^{-/-} and WT cells just by differentiation would show similar results. In addition, this point of view is even supported by a recent publication showing that deletion of NFATc1 increases the frequencies of memory T cells in vivo (Pachulec et al., 2016).

ESSENTIAL REVISIONS:

1) The comparison whether differences in cytotoxicity are due to the observed differences in the composition of subpopulations (Fig. 2e) is an important issue. To this end purified or gated subpopulations should be used from NFATc1^{-/-} and WT mice and compared.

2) To Fig. 2c and 2e:

Flow cytometry data should be presented in exemplary contour plots and overlays of histograms, too. MFIs should be presented (in particular for GrzmB and Prf1) in addition to % positive cells.

3) To Fig 2e:

Controls from uninfected mice should be provided.

4) Introduction/Discussion:

An introduction of differentiation of CD8 cells and the respective cytotoxic properties would be important to understand the context. Additionally, the discussion should be focused on this issue.

5) Results/Discussion:

The authors must quote and properly discuss the paper by Pachulec et al. as a recent reference with some overlapping and some opposite results (Pachulec et al., 2016).

6) Fig 2c:

The authors should provide NFATc2^{-/-} data for 2c like in all other figures.

8) page 9:

Tone down: 50% inhibition of IFN γ gene expression by NFATc2 ablation is not mild and shouldn't be ignored!

9) page 11:

Provide a reference if you state: The transcript level of almost all genes controlling aerobic glycolysis...

10) Results and discussion part:

Cytokine expression is an important feature and part of many figures. However, only IL-2 expression is a bit discussed. Please provide a comprehensive discussion about the knowledge and results in the light of different CD8 subpopulations.

MINOR POINTS:

1) Provide unique column identifications for Fig. 1a and b.

2) Correct one column description from 2e: CD62L⁺ CD44⁺ into CD62L⁻ CD44⁺.

3) To Fig. 3d,e:

It is difficult to oversee the conditions of the figure. I recommend including the identification for aCD3/CD28 and IL-2 treatments.

4) Provide a legend for Fig. 6g.

5) Provide names and MFIs for suppl. Fig. 2.

6) Correct the wrong characters in suppl. Fig. 3 and 7.

References:

Pachulec, E., Neitzke-Montinelli, V., and Viola, J.P. (2016). NFAT2 Regulates Generation of Innate-Like CD8⁺ T Lymphocytes and CD8⁺ T Lymphocytes Responses. *Front Immunol* 7, 411. doi: 10.3389/fimmu.2016.00411.

Wherry, E.J., and Ahmed, R. (2004). Memory CD8 T-cell differentiation during viral infection. *J Virol* 78, 5535-5545. doi: 10.1128/JVI.78.11.5535-5545.2004.

Reviewers' comments:

Reviewer #1 (Remarks to the Author):

This is a very interesting comprehensive study. For the most part the authors have addressed my initial questions.

Reviewer #2 (Remarks to the Author):

The authors now clarify which floxed mice and which *crd* mice have been used, and need to make sure that this is clearly stated in the manuscript and figure legends. I remain concerned about the use of CD4-Cre which ablates at double negative stage of thymic development, i.e. well before thymic selection is complete and single positive CD8 cells are produced. Thus the CD8 cells from the CD4-Cre deleted mice will have been skewed during thymic differentiation. The distal *Lck* promoter used in Figure 7 is much better as it deletes after positive selection. If I understand correctly the authors have used the CD4-Cre for most experiments and the *dLck*-Cre only for data in Figure 7. This makes the data from the figures difficult to compare and it is not possible to know whether changes seen in Fig1-6 are simply the result of disrupted thymic selection.

Reviewer #3 (Remarks to the Author):

1. Please consider the main concern adequately:

Each CD8 cell subpopulation (even among the memory cells) has a different strength of cytotoxic capability and other effector functions (Wherry and Ahmed, 2004). Different frequencies of memory T cells after differentiation of *NFATc1*^{-/-} and WT cells (as shown by Pachulec et al.) lead per se to different cytotoxic capabilities and cytokine expression (Pachulec et al., 2016).

The authors should comprehensively discuss this issue in the light of their results.

2. To Suppl. Fig. 3:

The unstimulated controls in the histograms (3c) are missing. The legend states *Nfatc2*^{-/-} but these data are not shown (3e).

References:

Pachulec, E., Neitzke-Montinelli, V., and Viola, J.P. (2016). NFAT2 Regulates Generation of Innate-Like CD8⁺ T Lymphocytes and CD8⁺ T Lymphocytes Responses. *Front Immunol* 7, 411. doi: 10.3389/fimmu.2016.00411.

Wherry, E.J., and Ahmed, R. (2004). Memory CD8 T-cell differentiation during viral infection. *J Virol* 78, 5535-5545. doi: 10.1128/JVI.78.11.5535-5545.2004.

Reviewer #1 (Remarks to the Author):

Nat Comms 16-28997

Dear Reviewer,

Please see our statements to your critical comments on our manuscript:

Comment 1:

1. While the finding is interesting the exact significance and mechanism surrounding the recruitment of mitochondria and cytotoxic granules to the IS in Figure 1E is unclear

In CD4⁺ lymphocytes, mitochondria are translocated to the IS following TCR activation (1). In addition Orai1 Ca²⁺ channels accumulate at the IS (2). The interplay between mitochondria and Orai1 channels at the IS is highly relevant for the generation of local (at the IS) and global Ca²⁺ signals (in the cytosol and in the nucleus). Mitochondria take up Ca²⁺ and reduce local Ca²⁺ domains at the IS, which in turn inhibit Ca²⁺ dependent inactivation of Orai1 channels (1,2). Thus mitochondria control local and global Ca²⁺ signals following TCR induced Ca²⁺ entry by Orai1 channels. By controlling Ca²⁺ signals, mitochondria regulate NFAT translocation into the nucleus in CD4⁺ lymphocytes (2). In the mast cell line RBL-1, it has been recently shown that local Ca²⁺ domains near Orai1 channels control NFAT1 translocation into the nucleus (3).

The mechanism of mitochondrial relocation to the IS involves cytoskeleton rotation of the MTOC (4). Interestingly, a recently developed theoretical model predicts that, in addition to rotation-driven relocation of the mitochondria attached to the MTOC and microtubules, an in homogenous distribution of plasma membrane ATPases with an accumulation at the IS controls global and local Ca²⁺ concentrations (4). In accordance with this prediction, an accumulation of plasma membrane ATPases has been reported in CD4⁺ lymphocytes (2).

In conclusion an impaired mitochondrial translocation of mitochondria to the IS should significantly decrease Ca²⁺ signals and T cell function including lytic (= cytotoxic) granule release.

Lytic granules are also transported to the IS in a MTOC-dependent way (5), probably by the same cytoskeletal rotation mechanism as relevant for mitochondria (4). Interestingly lytic granules are paired with vesicles transporting TCR signaling molecules like CD3 to the IS (6) highlighting the intimate coupling between TCR signaling and lytic granule release at the IS necessary for target cell killing.

Thus, by connecting specific metabolic needs (energy and calcium through mitochondria translocation) with specific functional needs (killing of target cells through lytic granule translocation), the local cellular subdomain close to the IS of CTL is well suited to work with high precision and efficiency.

1. Quintana A, Schwindling C, Wenning AS, Becherer U, Rettig J, Schwarz EC, Hoth M (2007). T cell activation requires mitochondrial translocation to the immunological synapse, *PNAS* 104, 14418-23.
2. Quintana A, Pasche M, Junker C, Al-Ansary D, Rieger H, Kummerow C, Nuñez L, Villalobos C, Meraner P, Becherer U, Rettig J, Niemeyer BA, Hoth M (2011) Calcium microdomains at the immunological synapse: how ORAI channels, mitochondria and calcium pumps generate local calcium signals for efficient T-cell activation. *EMBO J* 30, 3895-3912.
3. Kar P, Parekh AB (2016). Distinct spatial Ca²⁺ signatures selectively activate different NFAT transcription factor isoforms. *Mol Cell*. 2015 Apr 16;58(2):232-43. doi: 10.1016/j.molcel.2015.02.027.

4. Maccari I, Zhao R, Peglow M, Schwarz K, Hornak I, Pasche M, Quintana A, Hoth M, Qu B, Rieger H. (2016) Cytoskeleton rotation relocates mitochondria to the immunological synapse and increases calcium signals (2016). *Cell Calcium* 60, 309-321.
5. De la Roche M, Asano Y, Griffiths GM (2016). Origins of the cytolytic synapse. *Nat Rev Immunol* 16, 421-432.
6. Qu B, Pattu V, Junker C, Schwarz EC, Marshall M, Matti U, Becherer U, Bhat S, Kummerow C, Neumann F, Pfreundschuh M, Rieger H, Rettig J, Hoth M (2011). Docking of lytic granules at the immunological synapse in human CTL requires Vti1b-dependent pairing with TCR endosomes. *J Immunol* 186, 6894-6904.

A shortened version of this comment was introduced into the Discussion of the modified version of our MS (see page 18).

Comment 2:

2. Again the significance and mechanism of the generation of the different sizes of Nfat1c in Figure 3 is unclear

We improved now the labelling of various treatments of cells in Figs. 3d and e.

The rapid induction of short isoform NFATc1/ α A by TCR signals (its band is labelled by an asterisk in Fig. 3d) distinguishes the induction of NFATc1 from that of NFATc2 by immune receptor signals in primary (and other) lymphocytes. In the Discussion (on pages 15/16) we extended this fact and our hypothesis on the “two genes” in NFATc1

The lower blot in Fig. 3d which was performed with an Ab raised against the N-terminal α -peptide of NFATc1/ α A (see also Fig. 3b for its localization) indicates the appearance of NFATc1/ α A. Unfortunately, upon TPA+ionomycin induction of CTLs (lane 12; 8+) NFATc1/ α A was often degraded – probably due to high levels of proteases in CTLs.

Comment 3:

3. Throughout the paper the authors keep referring to genes controlling aerobic glycolysis this is wrong. These are simply genes involved in glycolysis.

In the new version of the MS we corrected this “faux pas”.

Comment 4:

4. Many of the findings can be controlled by mTOR activity. Have the authors examined mTOR activity in the knockout cells? Also, have they examined the role of Nfatc1 on Slc7a5 which is a calcineurin dependent leucine transporter?

Regarding **mTOR activity**: In Supplementary Fig. 9b we included now immune blots showing the effect of NFATc1 and IL-2 on the phosphorylation of mTOR target 4E-BP1. On page 13, last chapter, we added the following comments:

“The effect of NFATc1 ablation on glycolysis led to the question of whether NFATc1 exerts a direct effect on glycolytic genes. However, in CHIP seq assays apart from the *Hk2* and *Gapdh* genes (**Supplementary Fig. 10a**) no further genes of the glycolysis cascade showed distinct NFATc1 binding. One mediator of the

NFATc1 effect could be IL-2 whose expression is strongly diminished in *Nfatc1*^{-/-} T cells (**Fig. 5e**). IL-2 is known to activate multiple metabolic and transcriptional programs in T cells, mainly through the Ser/Thr kinase mTORC1, a prominent IL-2 target in T cells⁵. When we tested the effect of IL-2 in splenic CD8⁺T cells on 4E-BP1, a direct target of mTOR signals, we observed a marked increase in TCR-mediated phosphorylation that was inhibited by rapamycin. However, NFATc1 ablation was without effect on 4E-BP1 phosphorylation (**Supplementary Fig. 9b**). Nevertheless, adding 100 U/ml IL-2 to CD8⁺T cell cultures that were activated by α CD3/CD28 for 3 d revealed a marked increase in glycolysis of NFATc1-deficient cells (**Fig. 7g**). This finding is supported by similar RNA levels of glycolytic genes in WT and *Nfatc1*^{-/-} CTLs that were cultured in IL-2-containing medium for 6 d. Although we observed a drop in *Slc2a1* and *Hk2* RNA levels, we observed a similar glucose uptake and glycolysis of these cells (**Supplementary Figs. 9c-f**). Even upon re-stimulation of CTL- cells by α CD3/CD28 for 16 h, no differences in glycolysis and glycolytic capacity were detected between both types of cells (**Supplementary Fig. 9f**).”

The citation no. 5 (no. 31 in the MS) is the novel paper by S.H. Ross et al., 2016⁵.

Regarding **Slc7a5**: we show now in **Supplementary Fig. 10b** the binding of NFATc1 to the *Slc7a5* gene in ChIP seq assays, and added on page 14 the following comment:

“These findings do not exclude that NFATc1 affects also the metabolism of CTLs. The strong NFATc1 binding to the *Slc7a5* gene (**Supplementary Fig. 10b**) encoding a leucine transporter that coordinates the metabolic reprogramming of CTLs⁶ suggests that in CTLs NFATc1 controls metabolic pathways other than glycolysis”.

Comment 5:

5. I disagree with the comments in the second paragraph of the discussion regarding anergy. If the authors insist on this view they must defend it better. Also in the discussion they overstate the role of their findings on metabolism. This should be tempered by the fact that there are multiple inputs to metabolic reprogramming. This includes the concept of metabolic syndrome for CD8+ T cells which I think is not developed well enough to include in this work.

We extended now our statement regarding anergy as follows (in the 2. Paragraph of the Discussion, on page 15) :

“In CTLs, NFATc1 and NFATc2 overlap in their binding to almost all genes (and sites) but differ strongly in their transcriptional activity (**Figs. 4 and 5**). These data seem to contradict published data on the role of NFATs, in particular of NFATc2 (NFAT1), in the exhaustion of CD8⁺T cells⁷, an anergy-like state of activated CD8⁺T cells chronically infected with viruses. However, they are not necessarily in conflict with a promoting role of NFATs in CD8⁺T cell exhaustion. NFATc1 and NFATc2 differ in their expression in activated lymphocytes. While short immune receptor signals trigger the rapid translocation of both factors into the nucleus, more persistent signals induce the transcription of *Nfatc1* gene leading to the massive synthesis of NFATc1/ α A within a few hours (**Fig. 3d**). Although within several days the levels of

NFATc2 also increase (**Fig. 3e**), signals other than TCR triggering result in the further increase of NFATc2 in peripheral T cells⁸. These and further data led us to conclude that the *Nfatc1* gene codes for (at least) two factors which differ remarkably in their transcriptional properties². In our view (i) NFATc1/ α A which is highly expressed in activated T cells (**Fig. 3d**) supports ‘immunity’, i.e. the effector function of lymphocytes, including glycolysis, whereas (ii) NFATc1/ β C (and other longer NFATc1 isoforms) which is mainly expressed in resting T cells, supports the induction of ‘tolerance’ and ‘exhaustion’ of lymphocytes². The latter shares peculiarities with NFATc2, such as two SUMOylation sites within the C-terminal domain that is missing in NFATc1/ α A, and SUMOylation converts NFATc1/ β C to a suppressor of IL-2 induction⁹.”

In our view, the main finding of our study that NFATc1, i.e. NFATc1/ α A (see Fig. 7f or its predominant expression in aCD8⁺T cells: Fig. 3), but not NFATc2 controls glycolysis supports well the view that NFATc1/ α A controls immunity – and not anergy.

From the last page of Discussion (on page 17) we deleted the remark on the “metabolic syndrome”. We fully agree with your statement that in addition to glycolysis there are multiple inputs to the metabolic reprogramming of CD8⁺T cells.

Finally, let us thank for your helpful critical comments that certainly helped to improve the quality of our manuscript.

Edgar Serfling

Further References:

- 1 Serfling, E., Chuvpilo, S., Liu, J., Hofer, T. & Palmethofer, A. NFATc1 autoregulation: a crucial step for cell-fate determination. *Trends Immunol* **27**, 461-469, doi:S1471-4906(06)00243-2 [pii] 10.1016/j.it.2006.08.005 (2006).
- 2 Serfling, E. *et al.* NFATc1/ α A: The other Face of NFAT Factors in Lymphocytes. *Cell Commun Signal* **10**, 16, doi:10.1186/1478-811X-10-16 [pii] (2012).
- 3 Hock, M. *et al.* NFATc1 induction in peripheral T and B lymphocytes. *J Immunol* **190**, 2345-2353, doi:10.4049/jimmunol.1201591 [pii] (2013).
- 4 Muhammad, K. *et al.* NF- κ B factors control the induction of NFATc1 in B lymphocytes. *Eur J Immunol* **44**, 3392-3402, doi:10.1002/eji.201444756 (2014).
- 5 Ross, S. H. *et al.* Phosphoproteomic Analyses of Interleukin 2 Signaling Reveal Integrated JAK Kinase-Dependent and -Independent Networks in CD8(+) T Cells. *Immunity* **45**, 685-700, doi:10.1016/j.immuni.2016.07.022 (2016).
- 6 Sinclair, L. V. *et al.* Control of amino-acid transport by antigen receptors coordinates the metabolic reprogramming essential for T cell differentiation. *Nat Immunol* **14**, 500-508, doi:10.1038/ni.2556 [pii] (2013).
- 7 Martinez, G. J. *et al.* The transcription factor NFAT promotes exhaustion of activated CD8(+) T cells. *Immunity* **42**, 265-278, doi:10.1016/j.immuni.2015.01.006 S1074-7613(15)00032-1 [pii] (2015).

- 8 Hukelmann, J. L. *et al.* The cytotoxic T cell proteome and its shaping by the kinase mTOR. *Nat Immunol*, doi:10.1038/ni.3314 [pii] (2015).
- 9 Nayak, A. *et al.* Sumoylation of the transcription factor NFATc1 leads to its subnuclear relocalization and interleukin-2 repression by histone deacetylase. *J Biol Chem* **284**, 10935-10946, doi:10.1074/jbc.M900465200 [pii] (2009).

Reviewer #2 (Remarks to the Author):

Nat Comms 16-28997

Dear Reviewer,

Please see our statements to your critical comments on our manuscript:

1. Comment:

The methods section describes two genetic models for NFATc1 deletion, using a floxed NFATc1 mouse which is stated to be described in citation 46 (Lee et al “A critical role for Dnmt1 and DNA methylation...”) This does not seem to describe the generation of a floxed NFATc1 mouse, did they perhaps mean reference 47? What is really needed is the primary citation describing the exact construct for NFATc1 locus floxing (which neither citation provide). In the acknowledgements the authors mention that Anjana Rao has provided floxed mice “before publication.” It is critical to know exactly what is being deleted for interpretation of the data. Without this information this paper cannot be properly reviewed.

The majority of our assays were performed with mice bearing *Nfatc1^{flx}* alleles that we received more than 10 years ago from Anjana Rao’s laboratory before publication. Our first approach using those mice was published in 2011 (with two of A. Rao’s co-workers as co-authors: Edward D. Lamperti & Martin R. Müller). In this paper, we showed the effect of NFATc1 ablation on B cells. In the Supplement of the paper, we showed the complete inactivation of NFATc1 expression in splenic B cells of *Nfatc1^{flx/flx} x mb1-cre* mice¹⁰. There, in the Materials & Methods section, we mentioned that the generation of those mice has “to be described in detail elsewhere”. The first detailed description of the targeting strategy and the targeting vector were published in our paper by Vaeth et al. (2012)¹¹ (Ref. 47 in our MS). In several Supplementary figures of this paper, we showed (1.) the loss of NFATc1 expression in peripheral CD4⁺T cells from *Nfatc1^{fl/fl} x CD4-cre* mice [Figure S1C], (2.) the loss of NFATc1 expression in DP and CD8SP thymocytes [Fig. S1D], and (3.) the quite normal development of T cells in adult *Nfatc1^{fl/fl} x CD4-cre* [Figs. S6C] and newborn *Nfatc2^{-/-} x Nfatc1^{fl/fl} x CD4-cre* double-deficient mice [S7], respectively.

Later on, the targeting strategy for those mice with *Nfatc1^{flx}* alleles was also published by A. Rao’s laboratory (2015)⁷ (in Fig. S5 of this paper) – identical to the Fig. S1 in our paper¹¹ (see Figure below).

We have to apologize for a mistake in our citations: instead of citing the publication by Oh-hara et al. (Ref. 48 in our MS) we had to cite Martinez et al. (2015)⁷. This has been corrected now in the new version of the MS.

In Ref. 46 (as cited in our MS) the *CD4-cre* mouse line was described for the first time¹². In Ref. 49 (now Ref. 48), the *Nfatc1^{flx/flx}* line generated in L. Glimcher's laboratory was described. We used such mice in a few assays but no differences (e.g. in proliferation and AICD) were detected between T cells from the two *Nfatc1^{flx/flx}* lines.

2. Comment:

The next key issue is which Cre was used to generate the mice used in each experiment and how efficient was the excision in each case? Was any protein produced in any of the cells used? Although the investigators have antibodies to different splice products from NFATc1 there is almost no data to show loss of the protein.

For your inspection: Part of the *Nfatc1* locus and the targeting vector for the generation of *Nfatc1^{flx}* alleles^{11, 7}.

It is important to know which Cre was used as the CD4 Cre will be activated during the double-positive stage of thymocyte development and may therefore cause defects in the development of mature CD8/CTL. The dLck-Cre comes on after positive selection of thymocytes and is therefore more appropriate for studying the role of NFATc1 in CTL. +However, the efficiency of excision has been found to vary according to the floxed gene and it is vital to show efficiency of excision. Furthermore, use of this Cre is only described for Figure 7. Does this mean that all other experiments used the CD4-Cre? All of these issues need to be resolved before the paper can be properly assessed.

In the majority of our assays, we used *CD4-cre* mice¹², and in immune blots using protein extracts from activated CD8⁺Ts or CTLs cells from *Nfatc1^{fl/fl}* x *CD4-cre* mice

we did not observe any NFATc1 expression (see the novel Supplementary Fig. 4b in the modified version of the MS). In the assays of Fig. 7f, we used *Nfatc1^{flx/flx} x dlck-cre* mice. As shown now in the novel version of the Fig. 7f, any cytosolic and nuclear NFATc1 expression in activated CD8⁺T cells (lanes 1+2) was abolished (lanes 3+4), and due to cre activity the c.n.NFATc1/αA protein was properly expressed from the Rosa26 locus allele (see lanes 5+6 of Fig. 7f).

3. Comment:

One other aspect that the authors may wish to consider is that at the moment the manuscript comes across as a collection of studies from a number of investigators that have been strung together in which the quality of the data varies. For example in Figure 1 the imaging in a-d is excellent, but the resolution in e is so greatly reduced that the statements regarding polarisation of lytic granules and mitochondria are not supported. Mitotracker and lysotracker should image discrete puncta within the cytoplasm rather than the diffuse signals shown.

Due to your criticism, we performed a set of novel short-term stimulations of WT and *Nfatc1^{-/-}* CTLs with αCD3/CD28-coated beads and mitotracker and lysotracker stainings. The results of these novel assays – replacing now the former stainings in Fig. 1e - support our former results (as mentioned in the text) but are much sharper and show more details for the quite round lytic granules (in red) and the mitochondria (ingreen).

References:

- 1 Serfling, E., Chuvpilo, S., Liu, J., Hofer, T. & Palmetshofer, A. NFATc1 autoregulation: a crucial step for cell-fate determination. *Trends Immunol* **27**, 461-469, doi:S1471-4906(06)00243-2 [pii] 10.1016/j.it.2006.08.005 (2006).
- 2 Serfling, E. *et al.* NFATc1/alphaA: The other Face of NFAT Factors in Lymphocytes. *Cell Commun Signal* **10**, 16, doi:10.1186/1478-811X-10-16 [pii] (2012).
- 3 Hock, M. *et al.* NFATc1 induction in peripheral T and B lymphocytes. *J Immunol* **190**, 2345-2353, doi:10.4049/jimmunol.1201591 [pii] (2013).
- 4 Muhammad, K. *et al.* NF-kappaB factors control the induction of NFATc1 in B lymphocytes. *Eur J Immunol* **44**, 3392-3402, doi:10.1002/eji.201444756 (2014).
- 5 Ross, S. H. *et al.* Phosphoproteomic Analyses of Interleukin 2 Signaling Reveal Integrated JAK Kinase-Dependent and -Independent Networks in CD8(+) T Cells. *Immunity* **45**, 685-700, doi:10.1016/j.immuni.2016.07.022 (2016).
- 6 Sinclair, L. V. *et al.* Control of amino-acid transport by antigen receptors coordinates the metabolic reprogramming essential for T cell differentiation. *Nat Immunol* **14**, 500-508, doi:10.1038/ni.2556 [pii] (2013).
- 7 Martinez, G. J. *et al.* The transcription factor NFAT promotes exhaustion of activated CD8(+) T cells. *Immunity* **42**, 265-278, doi:10.1016/j.immuni.2015.01.006 S1074-7613(15)00032-1 [pii] (2015).
- 8 Hukelmann, J. L. *et al.* The cytotoxic T cell proteome and its shaping by the kinase mTOR. *Nat Immunol*, doi:10.1038/ni.3314 [pii] (2015).
- 9 Nayak, A. *et al.* Sumoylation of the transcription factor NFATc1 leads to its subnuclear relocalization and interleukin-2 repression by histone deacetylase. *J Biol Chem* **284**, 10935-10946, doi:10.1074/jbc.M900465200 [pii] (2009).
- 10 Bhattacharyya, S. *et al.* NFATc1 affects mouse splenic B cell function by controlling the calcineurin--NFAT signaling network. *J Exp Med* **208**, 823-839, doi:10.1084/jem.20100945 [pii] (2011).
- 11 Vaeth, M. *et al.* Dependence on nuclear factor of activated T-cells (NFAT) levels discriminates conventional T cells from Foxp3+ regulatory T cells. *Proc Natl Acad Sci U S A* **109**, 16258-16263, doi:10.1073/pnas.1203870109 [pii] (2012).
- 12 Lee, P. P. *et al.* A critical role for Dnmt1 and DNA methylation in T cell development, function, and survival. *Immunity* **15**, 763-774, doi:S1074-7613(01)00227-8 [pii] (2001).
- 13 Kaech, S. M. & Cui, W. Transcriptional control of effector and memory CD8+ T cell differentiation. *Nat Rev Immunol* **12**, 749-761, doi:10.1038/nri3307 [pii] (2012).

- 14 Sallusto, F., Geginat, J. & Lanzavecchia, A. Central memory and effector memory T cell subsets: function, generation, and maintenance. *Annu Rev Immunol* **22**, 745-763, doi:10.1146/annurev.immunol.22.012703.104702 (2004).
- 15 Wang, R. *et al.* The transcription factor Myc controls metabolic reprogramming upon T lymphocyte activation. *Immunity* **35**, 871-882, doi:10.1016/j.immuni.2011.09.021 S1074-7613(11)00515-2 [pii] (2011).

Finally, let us thank for your helpful critical comments that certainly helped to improve the quality of our manuscript.

Edgar Serfling

Reviewer #3 (Remarks to the Author):

Nat Comms 16-28997

Dear Reviewer,

Please see our statements to your critical comments on our manuscript:

SUMMARY:

The authors studied the role of NFATc1 in controlling effector functions of CD8+T cells. In particular, they analyzed in vitro differentiated CD8+ cells and compared NFATc1-/- and WT cells concerning cytotoxic effects, cytokine production, expression of NFATs, global gene expression, proliferation, and metabolic switch behavior.

Their main conclusion (title and summary) is that the transcription factor NFATc1 controls the cytotoxicity of murine CTLs. However, they neither fully characterized the cultured CD8 cells for composition in each experiment nor did they experiments with isolated defined Teff, Tem or Tcm cell subpopulations. This is a crucial point because each CD8 subpopulation has a different strength of cytotoxic capability and other effector functions (e.g. Fig 1. of Wherry and Ahmed, 2004). Therefore, differences of NFATc1-/- and WT cells just by differentiation would show similar results. In addition, this point of view is even supported by a recent publication showing that deletion of NFATc1 increases the frequencies of memory T cells in vivo (Pachulec et al., 2016).

ESSENTIAL REVISIONS:

Comment 1:

1) The comparison whether differences in cytotoxicity are due to the observed differences in the composition of subpopulations (Fig. 2e) is an important issue. To this end purified or gated subpopulations should be used from NFATc1-/- and WT mice and compared.

We show now in Supplementary Fig. 3a the differences in cytotoxicity between sorted CD62L⁺CD44⁺ and CD62L⁻CD44⁺ WT and *Nfatc1*^{-/-} CTLs, as you recommended.

Comment 2:

2) To Fig. 2c and 2e:

Flow cytometry data should be presented in exemplary contour plots and overlays of

histograms, too. MFIs should be presented (in particular for GrzmB and Prf1) in addition to % positive cells.

The data of those assays are now shown in the Supplementary Figs. 3c, d and e.

Comment 3:

3) To Fig 2e:

Controls from uninfected mice should be provided.

Those data from uninfected mice are now shown in the novel Fig. 2e (as “Control”).

Comment 4:

4) Introduction/Discussion:

An introduction of differentiation of CD8 cells and the respective cytotoxic properties would be important to understand the context. Additionally, the discussion should be focused on this issue.

See the novel paragraphs on pages 15 and 16. We introduced now on page 3 of Introduction comments and two citations (Ref. ¹³, ¹⁴) on the differentiation of CD8⁺T cells to T_{EM}, T_{CM} and T_{RM} cells.

Comment 5:

5) Results/Discussion:

The authors must quote and properly discuss the paper by Pachulec et al. as a recent reference with some overlapping and some opposite results (Pachulec et al., 2016).

See novel paragraph on page 16 of Discussion in which we compare our data with those presented by Pachulec et al., 2016.

Comment 6:

6) Fig 2c:

The authors should provide NFATc2-/- data for 2c like in all other figures.

Such data are now provided in Fig. 2c of the modified version of our MS.

Comment 7:

7) page 9:

Tone down: 50% inhibition of IFNg gene expression by NFATc2 ablation is not mild and shouldn't be ignored!

We corrected this point now as follows (2. paragraph, page 10):

“Although NFATc2 binds to almost the same sites as NFATc1 within and around the *Il3*, *Ifng* and *Il2* genes, NFATc2 ablation resulted in very different effects on their expression in CTLs. While NFATc2 ablation did not affect *Il2* induction, it reduced *Ifng* induction to approximately 55% and inhibited almost completely *Il3* induction (Fig. 5e).”

Comment 8:

8) page 11:

Provide a reference if you state: The transcript level of almost all genes controlling aerobic glycolysis...

We cited now the publication by Wang et al., 2011¹⁵ (page 12, 9th line from above, Ref. 27) as an excellent metabolism paper from which we adapted our scheme in Supplementary Fig. 8. Since, however, this is “textbook knowledge” and known since Warburg’s times, we hesitated to cite here any recent publication.

Comment 9:

9) Results and discussion part:

Cytokine expression is an important feature and part of many figures. However, only IL-2 expression is a bit discussed. Please provide a comprehensive discussion about the knowledge and results in the light of different CD8 subpopulations.

We extended now the discussions on cytokines (IL-3, IFN γ) on page 17(2. paragraph).

MINOR POINTS:

1) Provide unique column identifications for Fig. 1a and b.

In Fig. 1b we present now also a column compilation of data of Figs. 1a und 1c.

2) Correct one column description from 2e: CD62L+ CD44+ into CD62L- CD44+.

We are sorry for the mistake and corrected now the error.

3) To Fig. 3d,e:

It is difficult to oversee the conditions of the figure. I recommend including the identification for aCD3/CD28 and IL-2 treatments.

As you recommended we labelled now in Fig. 3d and e the lanes for aCD3/CD28 and IL-2 treatments.

4) Provide a legend for Fig. 6g.

In our “very latest manuscript” we found the following legend to Fig. 6g:

(g) Real time PCR assays of *Ccr7*, *IL7r*, *Itgae* and *Irf4* RNA expression in CTLs, normalized by Actb and relative to naïve CD8⁺T cells.

Since we presented the corresponding primers (in Table 1 of the Supplement) and the PCR technology in the section “Methods”, we thought that this short statement should be sufficient.

5) Provide names and MFIs for suppl. Fig. 2.

Names and MFIs are now provided for Supplementary Figs. 3c-e (former Supplementary Fig. 2).

6) Correct the wrong characters in suppl. Fig. 3 and 7.

These corrections have been done.

Finally, let us thank for your helpful critical comments that certainly helped to improve the quality of our manuscript.

Edgar Serfling

References:

- 1 Kaech, S. M. & Cui, W. Transcriptional control of effector and memory CD8+ T cell differentiation. *Nat Rev Immunol* **12**, 749-761, doi:10.1038/nri3307 [pii] (2012).
- 2 Sallusto, F., Geginat, J. & Lanzavecchia, A. Central memory and effector memory T cell subsets: function, generation, and maintenance. *Annu Rev Immunol* **22**, 745-763, doi:10.1146/annurev.immunol.22.012703.104702 (2004).
- 3 Wang, R. *et al.* The transcription factor Myc controls metabolic reprogramming upon T lymphocyte activation. *Immunity* **35**, 871-882, doi:10.1016/j.immuni.2011.09.021 S1074-7613(11)00515-2 [pii] (2011).

Reviewer #1:

This is a very interesting comprehensive study. For the most part the authors have addressed my initial questions.

Dear Reviewer,

We thank you for your valuable comments during the review process.

Reviewer #2 :

The authors now clarify which floxed mice and which crd mice have been used, and need to make sure that this is clearly stated in the manuscript and figure legends. I remain concerned about the use of CD4-Cre which ablates at double negative stage of thymic development, i.e. well before thymic selection is complete and single positive CD8 cells are produced. Thus the CD8 cells from the CD4-Cre deleted mice will have been skewed during thymic differentiation. The distal Lck promoter used in Figure 7 is much better as it deletes after positive selection. If I understand correctly the authors have used the CD4-Cre for most experiments and the dLck-Cre only for data in Figure 7. This makes the data from the figures difficult to compare and it is not possible to know whether changes seen in Fig1-6 are simply the result of disrupted thymic selection

Dear Reviewer,

We knew that the ablation of NFATc1 in Nfatc1 fl/fl x CD4-cre mice has no (or a minimal) effect on thymic selection of CD4+ and CD8+ T cells. The normal development of T cells in adult Nfatc1fl/fl x CD4-cre mice has been shown in Fig. S6C of our PNAS paper in 2012 (Ref. 1). The excellent penetrance of CD4-cre mediated inactivation of floxed genes prompted us to work with this mouse cre line, instead of using inducible cre or other lines.

Numerous, if not the majority of researchers in the field used CD4-cre lines to study CD8+T cells. This morning we discussed in our Journal Club the latest paper by Tak Mak (!, one of the fathers of k.o that just appeared 2017 in Immunity and in which CD4-cre mice were used to inactivate the Gclc gene (encoding the catalytic subunit of glutamate cysteine ligase [Ref. 2]) in CD4+ and CD8+ T cells. Further papers in which CD4-cre mice were used for studying CD8+T cells were published in Nature Immunity for measuring the “Control of amino-acid transport...” (by Sinclair et al., 2013 [3]) and the “Asymmetric inheritance of mTORC1 kinase activity...” (by Pollizzi et al. 2016 [4]), in Cell Reports on the “Mammalian target of rapamycin complex 2...” (by Zhang et al., 2016 [5]), in Immunity on “The transcription factor NFAT promotes exhaustion of activated CD8+ T cells” (by Martinez et al. , 2015 [6]) and in Frontiers in Immunology on “NFAT2 regulates generation.....” (by Pachulec et al., 2016 [7]).

Finally, I should also add that in the meantime several of the data we show in our MS (e.g. in Fig. 1a, actin defects and Fig. 6a, proliferation of CD8+T cells) were also performed with Nfatc1f/f x dlck-cre mice that we used in Fig. 7f (where we show that expression of the short NFATc1/αA protein enhances glycolysis). And similar results were obtained. Moreover, not all of data in Figs. 1-6 were elaborated using CD4-cre mice, such as those of Figs. 3 (Expression of NFATc1 in WT mice) and the majority

in Fig. 5 (ChIP seq assays using our tg mice expressing chimeric NFATc1/A-bio protein).

We thank you for your valuable comments during the review process. A comment on the use of CD4-cre mice is now presented in the Discussion (see pages 16/17, labelled in red).

References:

1. Vaeth, M., et al. Dependence on nuclear factor of activated T-cells (NFAT) levels discriminates conventional T cells from Foxp3⁺ regulatory T cells. *Proc Natl Acad Sci U S A* 109, 16258-16263 (2012).
2. Mak, T.W., et al. Glutathione Primes T Cell Metabolism for Inflammation. *Immunity* 46, 675-689 (2017).
3. Sinclair, L.V., et al. Control of amino-acid transport by antigen receptors coordinates the metabolic reprogramming essential for T cell differentiation. *Nat Immunol* 14, 500-508 (2013).
4. Pollizzi, K.N., et al. Asymmetric inheritance of mTORC1 kinase activity during division dictates CD8(+) T cell differentiation. *Nat Immunol* 17, 704-711 (2016).
5. Zhang, L., et al. Mammalian Target of Rapamycin Complex 2 Controls CD8 T Cell Memory Differentiation in a Foxo1-Dependent Manner. *Cell Rep* 14, 1206-1217 (2016).
6. Martinez, G.J., et al. The transcription factor NFAT promotes exhaustion of activated CD8(+) T cells. *Immunity* 42, 265-278 (2015).
7. Pachulec, E., Neitzke-Montinelli, V. & Viola, J.P. NFAT2 Regulates Generation of Innate-Like CD8⁺ T Lymphocytes and CD8⁺ T Lymphocytes Responses. *Front Immunol* 7, 411 (2016).

Reviewer #3 (Remarks to the Author):

Dear Reviewer,

Please see our statements to your critical comments on our manuscript:

Comment 1:

1. Please consider the main concern adequately:

Each CD8 cell subpopulation (even among the memory cells) has a different strength of cytotoxic capability and other effector functions (Wherry and Ahmed, 2004).

Different frequencies of memory T cells after differentiation of NFATc1^{-/-} and WT cells (as shown by Pachulec et al.) lead per se to different cytotoxic capabilities and cytokine expression (Pachulec et al., 2016).

The authors should comprehensively discuss this issue in the light of their results.

Response:

We have now extended the corresponding section on page 16 of Discussion (labelled in red) and cite both papers which you mentioned above.

Comment 2:

2. To Suppl. Fig. 3:

The unstimulated controls in the histograms (3c) are missing. The legend states Nfatc2^{-/-} but these data are not shown (3e).

Response:

We have now included the negative controls, and the legend of Fig. 3e has been corrected.

Finally, we would like to thank you for your valuable comments during the review process of our manuscript.

Edgar Serfling